# African Swine Fever Virus: A Review

**DOI:** 10.3390/life12081255

**Published:** 2022-08-17

**Authors:** Zhaoyao Li, Wenxian Chen, Zilong Qiu, Yuwan Li, Jindai Fan, Keke Wu, Xiaowen Li, Mingqiu Zhao, Hongxing Ding, Shuangqi Fan, Jinding Chen

**Affiliations:** 1College of Veterinary Medicine, South China Agricultural University, Guangzhou 510642, China; lzhaoyao123@163.com (Z.L.); chwenxian0912@163.com (W.C.); qiuzilong@stu.scau.edu.cn (Z.Q.); lyw13253374768@163.com (Y.L.); fanjindai@stu.scau.edu.cn (J.F.); 13660662837@163.com (K.W.); 18306616234@163.com (X.L.); zmingqiu@scau.edu.cn (M.Z.); dinghx@scau.edu.cn (H.D.); 2Guangdong Laboratory for Lingnan Modern Agriculture, Guangzhou 510642, China; 3Key Laboratory of Zoonosis Prevention and Control of Guangdong Province, Guangzhou 510642, China

**Keywords:** African swine fever, African swine fever virus, replication, virulence genes, vaccines, control, diagnosis

## Abstract

African swine fever (ASF) is a viral disease with a high fatality rate in both domestic pigs and wild boars. ASF has greatly challenged pig-raising countries and also negatively impacted regional and national trade of pork products. To date, ASF has spread throughout Africa, Europe, and Asia. The development of safe and effective ASF vaccines is urgently required for the control of ASF outbreaks. The ASF virus (ASFV), the causative agent of ASF, has a large genome and a complex structure. The functions of nearly half of its viral genes still remain to be explored. Knowledge on the structure and function of ASFV proteins, the mechanism underlying ASFV infection and immunity, and the identification of major immunogenicity genes will contribute to the development of an ASF vaccine. In this context, this paper reviews the available knowledge on the structure, replication, protein function, virulence genes, immune evasion, inactivation, vaccines, control, and diagnosis of ASFV.

## 1. Introduction

African swine fever (ASF) was first discovered in Kenya in 1921 and initially occurred in sub-Saharan African countries [1]. ASF is a severe infectious disease in pigs, caused by infection with the African swine fever virus (ASFV), and shows mortality rates close to 100%. ASFV genotype I was first introduced to Europe from western Africa in 1960 [2]. In 2007, ASFV type II was introduced from eastern Africa and spread widely across Europe, and in 2018, the virus was introduced to China via Russia [3]. On 3 August 2018, the World Organization for Animal Health (OIE) reported the first case of ASFV infection in Shenyang, Liaoning Province, China [4]. The strain belongs to genotype II [5]. Despite the emergency measures taken by the Chinese government, the subsequent outbreak of ASFV spread rapidly to 31 provinces and cities across the country. The clinical presentation and the gross pathological lesions of ASF in domestic pigs may vary depending on the virulence of the virus isolate, the route, the dose of infection, and host characteristics [6]. ASFV isolates can be classified as highly virulent, moderately virulent, and low virulent. The clinical courses observed in ASF in domestic pigs can be described as hyperacute, acute, subacute, or chronic. Acute ASF: The clinical course is characterized by high fever, with temperatures of 40–42 °C, lethargy, anorexia, and inactivity [7]. Subacute ASF: This clinical form is usually observed in animals infected by moderately virulent isolates, with similar clinical signs as those observed in acute ASF, although normally less marked. Affected pigs show moderate to high fever, and the mortality rate ranges from 30 to 70%, with pigs dying at 7–20 days after infection. The vascular changes, mostly hemorrhages and oedema, in the subacute form of the disease can be more intense than the acute form [8]. Chronic ASF: This clinical form is characterized by multifocal necrosis in the skin and arthritis, growth retardation, emaciation, respiratory distress, and abortion [7]. Warthogs, however, do not show any clinical symptoms when infected with ASFV [9]. ASFV can survive in the natural environment for an extended time because of the complex structure of its particles and genome. Domestic pigs are easily infected with ASFV through sick pigs, contact with people and objects carrying the virus, and exposure to a virus-contaminated environment [10]. In one study, parameters such as symptoms, pathogenicity, distribution of the virus in tissues, humoral immune response, and dissemination of the virus by blood, oropharyngeal, and rectal routes were investigated. The Polish ASFV caused a case of rapidly developing fatal acute disease, while the Estonian ASFV caused acute to sub-acute infections. In contrast, animals infected with the ASFV from Latvia developed a more subtle, mild, or even subclinical disease. Oral excretion was sporadic or even absent in the attenuated group, whereas in animals that developed an acute or sub-acute form of ASF, oral excretion began at the same time the ASFV was detected in the blood, or even 3 days earlier, and persisted up to 22 days. Regardless of virulence, blood was the main route of transmission of ASFV, and infectious virus was isolated from persistently infected animals for at least 19 days in the attenuated group and up to 44 days in the group of moderate virulence. Rectal excretion was limited to the acute phase of infection. In terms of diagnostics, the ASFV genome was detected in contact pigs in oropharyngeal samples earlier than in blood, independently of virulence [11]. In fact, the ASF outbreak has greatly impacted the global pig industry and many national economies [12]. Because of the powerful function of the viral genome and the lack of knowledge on proteins that may yield a protective immune response in pigs, the development of an ASF vaccine remains a worldwide challenge. Experimental live attenuated vaccine viruses, generated by deleting virus genes associated with virulence, have been shown to be effective [13,14,15,16,17,18]. One of these vaccines, ASFV-G-∆I177L [16], induces effective protection against its parental virulent virus strain, Georgia 2007, as well as a virulent Vietnamese field strain isolated in 2019 [19]. This vaccine has been selected for further development and commercialization, which includes a thorough assessment of its safety characteristics. Tran evaluated the safety profile of an efficacious live attenuated vaccine candidate, ASFV-G-∆I177L. Results from safety studies showed that ASFV-G-∆I177L remains genetically stable and phenotypically attenuated during a five-passage reversion to virulence study in domestic swine. In addition, large-scale experiments to detect virus shedding and transmission confirmed that even under varying conditions, ASFV-G-∆I177L is a safe live attenuated vaccine [20].

## 2. African Swine Fever

ASF is one of the most devastating diseases in domestic pigs and wild boars, causing high economic costs for the pork industry. The main route of ASF transmission includes the transportation of infected animals, and more commonly, the transportation of infected products. Infected wild boars are also a very important source of infection [21]. ASFV can also replicate in soft ticks of the genus *Ornithodoros*, including *O. moubata* in Africa and *O. erraticus* on the Iberian Peninsula, which are involved in the epidemiological cycles of ASF [22]. Although ASFV can be transmitted by some species of *Ornithodoros* soft ticks in endemic African areas, Pereira de Oliveira (2019) experimentally showed that Palearctic soft ticks, *O. erraticus* (from Portugal) and *O. verrucosus* (from Ukraine), were not able to transmit the current circulating European ASFV strain Georgia2007/1 [23]. The clinical course observed in the ASF of domestic pigs can be described as peracute, acute, subacute, or chronic. ASFV enters the body via the tonsils or dorsal pharyngeal mucosa and moves to the mandibular or retropharyngeal lymph nodes; from here, the virus spreads systemically through viremia. Thereafter, the virus is detectable in almost all pig tissues [24]. Clinical signs and lesions are induced by highly virulent ASFV strains and are characterized by high fever (i.e., a body temperature of 41–42 °C), loss of appetite, inactivity, dyspnea, and cutaneous hyperemia. Sometimes, sudden death can also be observed without signs of disease. Animals may show respiratory distress because of the high fever, but no gross lesions can usually be found during post-mortem examinations. Acute ASF is caused by highly or moderately toxic isolates. Its clinical course is characterized by high fever, a body temperature of 40–42 °C, lethargy, anorexia, inactivity, respiratory distress, and severe pulmonary edema. Animals affected by high pathogenicity [25] show skin lesions, presented with petechial hemorrhages or ecchymoses. Other clinical signs may include nasal discharge, vomiting, and diarrhea. Diarrhea causes the emergence of black stains in the perianal region of animals [6]. Pregnant sows may miscarry [26]. In most cases, the first introductions do not show high mortality nor characteristic clinical signs or lesions, but fever and some hemorrhagic lymph nodes [6]. At autopsy, the most characteristic lesion of acute ASF is severe hemorrhagic splenomegaly observed at the opening of the abdominal cavity of an animal with acute ASF. The liver is severely congested. Very large, dark-colored spleen with rounded edges (hemorrhagic splenomegaly), occupying a large volume of the abdominal cavity, were seen, along with multifocal hemorrhages in a lymph node with a marbled appearance, severe hemorrhagic lymphadenopathy in the gastrohepatic lymph node, severe hemorrhagic lymphadenopathy in the renal lymph node, severe hemorrhagic lymphadenopathy in the ileocecal lymph node, and moderate hemorrhagic lymphadenopathy in the mesenteric lymph node in (Figure 1) [27]. Subacute ASF shows similar symptoms to the clinical symptoms observed in acute ASF, although these are generally less pronounced. Infected pigs exhibit moderate to high levels of fever with a mortality rate between 30 and 70%; pigs commonly die 7–20 days after infection. Affected pigs show moderate to high fever, and the mortality rate ranges from 30 to 70%, with pigs dying at 7–20 days after infection. The vascular changes, mostly hemorrhages and oedema, in the subacute form of the disease can be more intense than in the acute form [8]. At the post-mortem examination, animals show hydropericardium, ascites, and multifocal oedema, very characteristic in the wall of the gall bladder or in the perirenal fat [6]. Some animals may show hemorrhagic splenomegaly as described for the acute form of the disease, but many animals will show partial splenomegaly, with patches of spleen affected and other areas unaffected. A multifocal hemorrhagic lymphadenitis can also be observed with multiple lymph nodes in all areas of the body showing the hemorrhages and the “marble” [28]. The clinical manifestations of chronic ASF are skin multifocal necrosis, arthritis, growth retardation, weight loss, dyspnea, and miscarriage (Table 1).

It has been reported that infection with the Kirawira isolate (KWH/12) in pigs leads to a decrease in leucocyte counts, while the red blood count remains almost unchanged [29]. In contrast, the total white blood cell counts did not show significant changes when lymphopenia and neutrophilia were observed [30]. With the progression of the disease, the number of neutrophils increased, while the number of lymphocytes decreased. A decrease in the lymphocyte numbers occurred between 2 and 4 days post-inoculation (DPI) and corresponded with a drop in the total WBC count. The minimum number of lymphocytes observed ranged from one-third to one-half of the pre-inoculation levels. A rise in neutrophil numbers was observed, gradually increasing from 2 to 3 DPI. This increase, which ranged from two to four times the pre-inoculation levels, was accounted for by an increase in juvenile forms. Before infection with ASFV, the difference in leukocyte count showed that the average concentration of lymphocytes was 61%, and the average concentration of neutrophils was 32%, while at the end of infection, the average concentration of lymphocytes and neutrophils were 32% and 64%, respectively [29]. One study used Georgia/2007 and Armenia/07 (genotype II) strains for animal experiments [31]. The percent of fifteen cell types based on observations of 600 total cell counts was calculated during ASFV infection. Marked changes in the percent of cells were observed from 1 dpi. From the early stages of infection, immature forms of white blood cells, such as lymphoblasts, monoblasts, and metamyelocytes, as well as atypical and reactive lymphocytes, appeared in the blood of infected pigs. The number of lymphoblasts reached its peak (18.9% of the total cells) at 5 dpi and sharply decreased in the final phase of infection. The number of atypical and reactive lymphocytes increased throughout the entire period of infection and reached its maximal value in the premortal stage. ASFV infection caused a decrease in the number of small- and medium-sized lymphoctyes, whereas the number of large-sized lymphocytes decreased but was not statistically significant. By the last day of infection, the percent of dead cells reached 24.5%, and the remaining cells were represented mainly by atypical and reactive lymphocytes, as well as monocytes and metamyelocytes (Table 2).

Ramiro-Ibanez et al., showed that, compared with pre-infection values, infection with highly virulent virus strains caused a reduction of 60% of macrophages and B-lymphocytes. After a short period, these are accompanied by increased B-lymphocyte counts (probably because of the polyclonal activation period of B-lymphocytes) and an increase in macrophages at the start of infection [32]. At the peak of viral titers, an increase was observed in both SLA II (MHC-II) and CD8+ expressions. The levels of both CD8+ and CD4+ T cells were elevated during the second week of infection [32]. Upon ASFV infection, the destruction of monocytes/macrophages causes the release of cellular components. Activated monocytes/macrophages secrete a wide range of mediators, including proinflammatory cytokines such as IL-1, IL-6, and TNF-α [33]. Comparison of moderately and highly virulent ASFV strains identified no general differences in cell tropism or organ distribution, but significantly more tissue is destroyed by strains with higher virulence. The observed cellular depletion was previously attributed to necrosis, but the changes were attributed to apoptotic mechanisms [28,34,35]. It is generally accepted that apoptosis of lymphocyte subsets, and impairment of hemostasias and immune functions caused by ASFV, can be attributed to cytokine-mediated interactions [36]. These are triggered by infected and activated monocytes and macrophages, rather than by virus-induced direct cell damage.

## 3. ASFV Structure

ASFV is a large, enveloped, double-stranded DNA (dsDNA) virus, the only member of the Asfarviridae family [37], and the only DNA virus transmitted by arthropod vectors [38]. Comparative genomic analysis grouped ASFV with nucleocytoplasmatic DNA viruses (NCLDVs), a distinct monophyletic group of viruses that infect a wide range of eukaryotic organisms [39]. NCLDVs including ASFV are reclassified into the new order ‘‘Megavirales’’ [40]. The NCLDVs superfamily contains: Poxviridae, Iridoviridae, Asfarviridae, Phycodnaviridae, Mimiviridae, Ascoviridae, and Marseilleviridae [39]. Although similar in structure, genome organization, and replication characteristics to other NCLDVs, the ASFV has unique features.

The structure of ASFV is an icosahedral multilayer structure, containing about 50 types of proteins, and the virus particles are 200 nm in diameter [41]. However, the diameter of purified virus particles in Chinese farms was 260–300 nm, which was identified through cryo-electron microscopy [42]. The virus contains a dsDNA genome with a length of 170–194 kb and includes inverted repeat sequences at the ends and a hairpin structure [43]. The virus genome contains more than 150 open reading frames [44] and encodes a variety of enzymes, including structural proteins and multiple enzymes involved in DNA replication, gene transcription, and protein modification [45]. Structural proteins include early RNA transcription and processing enzymes as well as factors, such as multi-subunit polymerase, PolyA polymerase, guanylate transferase, protein kinase, and inhibitor of apoptosis (IAP). At the same time, other enzymes are also active in the virus particle proteins, such as protein kinase [46], nucleoside triphosphate phosphohydrolase, acid phosphatase, and deoxyribonuclease [1]. The functions of almost half of the ASFV genes still remain unknown and await exploration [37].

Intracellular ASFV has a four-layer structure, a central genome-containing nucleoid wrapped by a protein core shell, an inner lipid envelope, and an icosahedral protein capsid (Figure 2) [47]. Mature virus particles acquire the outermost layer, called the envelope, as they emerge through the cell membrane [37]. Both intracellular mature virus particles and enveloped extracellular virus particles with vesicular membranes are infectious [48].

The nucleoid of ASFV is approximately 70–100 nm in diameter and is an electrodense structure, which contains the viral genome and associated proteins, such as viral structural protein p10 [47], pA104R, and parts of the transcriptional machinery. The nucleoid is assumed to contain proteins and enzymes that execute and assist gene transcription, including multisubunit RNA polymerases, polyadenylate polymerases, blocking enzyme, and early transcription factors [47]. The protein layer of the core shell has a thickness of about 30 nm and wraps around the nucleoid of the virus [50]. The core shell protein contains the polyprotein pp220 (p150, p37, p34, p14, and p5), pp62 (p35, p15, and p8), and pS273R [51]. The multiprotein pp220 plays a key role in core assembly, including the steps leading to genome wrapping, condensation, and viral nucleoprotein uptake, and p15 may be the major protein in this layer [45]. PS273R is an enzyme that processes viral polyproteins, assembles viruses, and catalyzes protein hydrolysis [45,52].

The inner envelope membrane is a single lipid membrane monolayer derived from the endoplasmic reticulum. This membrane contains the membrane proteins p54, p17, and H248R, as well as the viral attachment protein p12, which was previously assumed to be on the outer membrane [53]. Inner envelope protein 17 (D117L) is an essential and highly abundant protein that is required for capsid assembly and icosahedral morphogenesis [54]. P12 is associated with viral membrane precursors, assembly particles, and intracellular mature viruses [47]. A recent study showed that the viral capsid is composed of 2760 pseudo-hexametric capsomeres and 12 pentameric vesicles arranged in a triangular shape, forming the icosahedral morphology of the capsid [42]. This structure was also found in the capsid proteins of adenovirus, phage PRD1, and poxvirus. The capsomers on the capsid are organized into 20 threefold-axis-centered triangular arrays and 12 fivefold-axis-centered pentagonal arrays (Figure 3) [55]. Each pseudo-hexameric capsomer is composed of three p72 molecules, and five copies of the protein penton (a different protein than p72) constitute a pentameric capsomer [42]. This layer contains the major capsid protein p72 and the four minor capsid proteins M1249L, p17, p49, and H240R [42]. Protein p72, encoded by gene B646L, is the main component of the vesicle, and three p72 protein molecules form a double jelly roll structure to form pseudo-hexameric vesicle capsomeres [56]. B602L is required as a chaperone for the correct formation of protein p72. Twenty-four genotypes are based on the major coat protein p72, and eight serotypes are based on the viral hemagglutinins CD2-like protein (CD2v) as well as C-type lectin. All 24 ASFV genotypes have been identified in Africa [57]. The membrane structural protein p49 (B438L) is located near the capsid apex and is involved in the formation of the capsid acrosome [58]. PE12R is also localized near the capsid and is involved in membrane transport of viral particles. PE120R is required for virus transmission but not for viral infectivity [59]. Other known proteins on the capsid are M1249L and H240R [60].

The outer membrane or envelope is captured by viral particles as they bud through the cytoplasmic cell membrane. The viral attachment protein p12 is localized in the outer membrane. Protein p12 can interact with cellular protein receptors, thus promoting the attachment of ASFV to host cells [61]. Immunoelectron microscopy showed that proteins p24 (pkP177R), CD2v (pE402R), and pP2 (pO61R) are also located on this plasma membrane [62]. Protein p24 is present in the membranes of both uninfected and infected vero cells and is incorporated in the virus particles through the cell membrane, probably during the budding process [63]. Protein CD2v is considered to be a viral homologue of cellular protein [62].

## 4. Virus Replication

ASFV primarily infects macrophages. Notably, early studies of ASFV replication were performed in Vero cells. ASFV entry includes dynamin and clathrin-mediated endocytosis (CME) and micropinocytosis [64]. This diversity of pathways to enter host cells can also be found in other viruses [65]. In the early stages of ASFV infection, the lattice CME pathway is pH-dependent [66]. During endocytosis, the ASFV viral outer membrane breaks down under the acidic pH of late endosomes, and the viral core depends on transportation by the microtubule cytoskeleton to replication sites [64]. The cholesterol in the cell membrane is a necessity so that cells can be infected with the virus. Despite the proposed clathrin-dependent mechanism of cell entry, the specific receptors and attachment factors for viral entry into cells remain unknown [67]. Although no specific receptors for ASFV have been identified to date, several macrophage receptors are assumed to play a possible role, including CD163, CD45, MHC II, and a number of others (Table 3) [68]. Expression of CD45 is strongly associated with ASFV infection on adherent porcine bone marrow (PBM) cells. CD163 is a member of the scavenger receptor cysteine-rich domain family, which is a family of receptor cysteine-rich structural domains, the expression of which is restricted to the cells of monocytes/macrophages. Since CD163 expression is enhanced upon ASFV entry into cells, CD163 was considered to be the ASFV receptor; however, CD163 knockout pigs can still be infected by ASFV [68]. CD163 may be involved in ASFV infection but may not be required [68]. Mediated by the pE248R enzyme, the viral core is released into the cytoplasm. The viral core uses the ubiquitin-protease system to release viral genes [69].

ASFV uses its own encoded DNA replication enzymes (DNA topoisomerase, DNA helicases, polymerase, ligase, and binding protein) to initiate viral gene transcription immediately after infection, independent of cellular enzymes [75]. Viral genome replication begins in the nucleus, where small replication intermediates are produced and subsequently transferred to the cytoplasm; there, larger intermediates form and mature [76]. Virion morphogenesis takes place in a specific region of the cytoplasm near the nucleus and in the center of the microtubule tissue, a region called the viral factory [77]. This region is largely devoid of organelles but is surrounded by endoplasmic reticulum membranes in vimentin [78]. Approximately 20% of the ASFV genome is dedicated to the encoding, transcription, and modification of gene functions. DNA replication-associated enzymes are expressed immediately upon entry of the viral core particles into the cytoplasm [79]. Approximately 6 h post infection, viral DNA begins to replicate. The transcriptase packaged in the core of the virus initiates early mRNA transcription [80]. The viral polypeptide p54 is translationally inserted into the endoplasmic reticulum, and the modified endoplasmic reticulum membrane is converted into a membrane precursor within the virus factory [53]. The initial virus morphogenesis is the curved inner membrane precursor, and with progressing virus morphogenesis, the area the virus factory occupies in the cytoplasm increases [81]. The protein XP124L transfers the inner membrane precursor to the endoplasmic reticulum cavity, and protein XP124L can be detected by immunofluorescence electron microscopy in both ASFV production intermediates and mature viroids [82]. At the edge of the viral factory, apparently invaginated endoplasmic reticulum (ER) can be found [78]. The lumen of the endoplasmic reticulum and membrane markers with labeled protein disulfide bond isomerase (PDI) can be detected in the space of marginal zipper-like structures. Interestingly, the polyproteins pp220 and pp62 interact to form a core shell under the internal lipid envelope [50]. During virus formation, the non-structural protein pB602L catalyzes the gradual formation of the capsid structure by folding the capsid major protein p72 and the capsid apex protein p49 [58]. The process of capsid assembly on the viral membrane for the formation of a polyhedral form requires adenosine triphosphate and calcium [83]. In the virus factory, icosahedral particles can be observed, either with or without a central core of electron density. Nucleoid formation may be the last step of virus assembly [81]. Multiprotein pp62 is essential for the correct assembly and maturation of the viral core [84]. ASFV viral particles are transported to the cell surface by intracellular microtubule transport routine kinesin with capsid protein pE120R and leave the host cell by budding [85]. Extracellular viral particles thus acquire a plasma membrane layer. Salas et al. developed a model of the process of ASFV generation (Figure 4), which also identified the role of a number of ASFV proteins in the process of virus generation [47]. Although the replication process of ASFV has been studied a lot, there are still gaps in the specific details of the replication mechanism of ASFV, which require further research. The virulence genes of ASFV and their interaction with the host are still unclear. It is necessary to conduct a more in-depth analysis of the ASFV genotype through high-quality whole-genome sequencing to improve our understanding of the epidemiological tracing of this disease, including the evolutionary relationship and the phenotypic differences between different ASFV strains and the inducing factors related to genomic variation.

## 5. Virulence Genes

ASFV causes a wide range of clinical signs in domestic pigs, from acute infections with up to 100% mortality to subclinical or even non-clinical infections. Acute infection of domestic pigs is associated with anorexia, fever, nosebleeds, erythema and cyanosis of the skin, black stool, and, in some cases, diarrhea, with death occurring 12 to 14 days after infection [86]. Moderately virulent strains cause a subacute form of ASF, which causes milder clinical signs and injury than the acute form, with death occurring 15 to 20 days after infection [87]. The pathogenicity of ASF varies with the virulence or viral dose.

Nurmoja et al., concluded that oral and intranasal inoculation with the same strain resulted in acute and severe disease in wild boars, and only one animal was spared from infection [88]. Knowledge about the genome associated with the virulence of ASFV isolates is still lacking, and further research is required to understand the important components that cause this disease [89]. Current methods for determining changes in virulence and pathogenesis are based on classical experimental infections. A number of ASFV genes are associated with the virulence of the virus, but do not affect virus replication in macrophages in vitro. Past studies indicated that most of the genotype II isolates of the “Georgia 2007 type”, which is prevalent in Eastern and Central Europe and recently Asia, are highly virulent, resulting in very high mortality rates of 91–100% [90]. However, 2–10% of infected animals recover from acute ASFV infection. These survivors may develop persistent infection in tissues throughout the body and develop disease under certain predisposing conditions (e.g., transport, underfeeding, and immunosuppression) [5].

UK (DP96R) and 23-NL (DP71L or I14L) are two ASFV virulence genes adjacent to each other in the right variable region of the genome. UK is an ASFV early replication protein with no similarity to other known proteins [91]. Deletion of the UK gene from pathogenic ASFV does not affect virus growth in macrophages, but significantly attenuates ASFV in pigs [70]. The 23-NL gene encodes the NL protein, a protein similar to the herpes simplex virus neurovirulence factor ICP34.5. ICP34.5 plays a role in viral maturation and release. It mainly prevents host protein shutoff by directing dephosphorylation of eIF2α via protein phosphatase 1α (PP1) [92,93]. NL may play a similar role to ICP34.5 in ASFV infection, as deletion of 23-NL from ASFV strain E70 diminished its virulence in pigs without affecting virus replication in macrophages in vitro [94]. Genetic deletion of MGF genes on ASFV BA71 isolates reduced the ability of the virus to replicate in porcine macrophages [95,96]. Knockout of MGF360-12L, -13L, and -14L from the ASFV Pretorius/96/4 isolates reduced the ability of the virus to replicate in ticks [97]. Deletion of MF360-9L, -10L, -11L, -12L, -13L, and -14L together with MGF505-1R and -2R genes significantly reduced the ability of ASFV Pretorius/96/4 isolates to replicate in porcine macrophages. MGF360-12L, -13L, and -14L together with MGF505-1R, MGF505-2R, and MGF505-3R deletions completely attenuated the normally strong virulence [97]. Deletion of the TK gene in the ASFV-G/VP30 strain reduces virus replication in vero cells and does not cause morbidity in pigs [98]. The live attenuated ASFV developed by ASFV virulence-related genes (UK, 23-NL, TK, 9GL, or MGF) is protected when challenged by the homologous virulent parent virus [14,97,99,100]. Cackett et al. identified a total of 91 ASFV genes that existed with differential expression between early and late infection through transcriptomics analysis, of which 36 are early genes, leaving 55 classified as late genes [101]. These studies have laid the foundation to advance our knowledge on the expression and potential functions of ASFV genes, and for the development of antiviral drugs that target the gene expression machinery (Table 4).

## 6. Immunoescape

ASFV uses various mechanisms to escape the host immune system, such as type I interferon (IFN) response, apoptosis, inflammatory response, and activation of specific target genes (Table 5) [100]. ASFV inhibits expression of TNF-α, IFN-α, and IL-8 while inducing the production of TGF-β from infected macrophages [161]. Certain viral proteins have been reported to have the ability to regulate and inhibit programmed cell death pathways early during viral infection. Prominent examples are A179Lp, a nonessential protein that promotes viral replication and the production of offspring viruses, as well as Bcl2 family members, A224Lp, IAP family members, and EP153Rp, a C-type lectin. Notably, ASFV strains with different virulence phenotypes differ in their ability to induce expression of pro-inflammatory cytokines or interferon-related genes in macrophages in the early stages of infection [33]. A238L shows sequence similarity with IκB and prevents activation of nuclear factor kappa B (NF-κB)-dependent gene transcription. The similarity between A238L and IκB is limited to the central regions of the proteins, which contain three ankyr-in-like repeats. The NH_2_- and COOH-terminal regions of A238L are different to those of the cellular IκB proteins, indicating that A238L may function via a mechanism that differs from that of IκB [162]. In contrast, other essential proteins (e.g., the structural protein p54/EP153R) induce apoptosis during late viral infection to regulate both the production and release of viral particles. The protein DP71LP recruits protein phosphatase 1 of the dephosphorylation initiation factor eIF2 and restores globin synthesis [163]. Deletion of multiple genes in the multigene family MGF360 and MGF530/505 results in an increased induction of IFN and genes stimulated by IFN in infected macrophages [164]. This suggests that a number of members of these gene families suppress the production of IFN. In addition, transforming growth factor β (TGF-β) was not detected in porcine macrophages infected in vitro with multiple gene deletion strains of 360 (MGF360) and MGF530/505, but not tumor necrosis factor or interleukin-1; moreover, the response of infected macrophages to interferon-γ and lipopolysaccharide was inhibited [165]. I239Lp is a glycoprotein located on the surface of the host cell membrane and is the first ASFV protein that was found to inhibit the IFN response via the TOLL-like receptor 3 signaling pathway. The protein k205Rp is located in the cytoplasm and inhibits the activation of IFN-β; in addition, the protein A276Rp is also an inhibitor of IFN-αβ activation. Inhibition of IFN-αβ activation results in the suppression of hundreds of IFN-activated genes [91]. The protein D96Rp (or UK) is also a potential immune escape gene [18]. Other regulatory proteins such as hemagglutinin CD2V/E402Rp, which is located on the surface of extracellular viral particles, can inhibit lymphocyte activation [166]. MGF505-7R (A528R) inhibits the induction of IFN by repressing IRF3 and NF-κB transcription factors. This protein also inhibits type I and type II interferon signaling pathways. However, the mechanism by which MGF 505-7R A528R inhibits interferon induction has not been well described, and neither have other proteins with this function [163]. DP96R of ASFV China 2018/1 plays an important role in viral immune evasion by negatively regulating IFN expression and NF-κB signaling through inhibition of TBK1 and IKKβ [167]. During early infection, attenuated NH/P68 activates the c GAS-STING-IRF3 cascade, which induces STING phosphorylation and translocation through the c GAMP mechanism to activate TBK1 and IRF3 and produces high levels of β-interferon (IFN-β). The strongly virulent Armenia/07 ASFV regulates the c GAS-STING pathway; however, these mechanisms do not work when porcine macrophages are infected with attenuated virulent NH/P68 ASFV [168]. A major feature of acute ASFV disease is the induction of massive apoptosis of uninfected B and T lymphocytes in lymphoid tissue and blood. Both lymphatic failure in primary and secondary lymphoid organs and the death of infiltrating lymphocytes in non-lymphoid organs have been attributed to massive apoptosis of lymphocyte subpopulations. However, the mechanisms of lymphocyte apoptosis are poorly understood [169]. Infected cells may either secrete or present cell surface factors that induce apoptosis [170]. Although the role of specific viral proteins in viral infections is well understood, the mechanism of the body’s immune response to the ASFV virus remains unclear.

### 6.1. Apoptosis

The induction of apoptosis in infected cells is an important mechanism by which host cells limit viral replication. Activation of this process prevents the virus from completing its replication cycle, thereby decreasing the number of produced infectious daughter viruses. As with other viruses, ASFV infection of cells induces apoptosis because it induces caspase 3 activation (Figure 5) [178]. In pigs, ASFV infection leads to apoptosis of thymic lymphocytes and megakaryocytes in the body [179]. Apoptosis of ASFV-infected macrophages has also observed later, after in vitro infection [178]. Activation of apoptosis in infected cells can limit the completion of the replication cycle of the virus and thus reduce the production of offspring virus [180]. The expression of tumor necrosis factor-α was found to be increased after ASFV infection both in vitro and in vivo. TNF-α may play a key role in the pathogenesis of ASF, including changes in vascular permeability, coagulation, and induction of apoptosis in uninfected lymphocytes [181]. Caspases 3 and 9 were induced in vero cells 16 h post infection (hrp) with ASFV, and this induction increased with infection, showing higher levels at 48 hrp [182]. ASFV triggers apoptosis early in the infection process, and activation of caspase 3 after viral infection does requires neither viral protein synthesis nor DNA replication. It is likely that the activation was caused by the fusion of the ASFV membrane with the endosomal membrane or by the induction of viral decidualization [183]. Interestingly, the ASFV protein p54 induced caspase-3 activation and apoptosis when transfected into cells [175]. Throughout ASFV infection, the levels of host protein p53 (a protein central to cell survival and cell cycle regulation) and host protein Bax (a pro-apoptotic protein) are upregulated. This suggests that the p53-dependent apoptotic pathway may play a role in the apoptosis of ASFV-infected cells [174]. As an endoplasmic reticulum stress virus that induces elevated levels of Hsp60 in infected cells, ASFV is predicted to activate the pro-apoptotic transcription factor CHOP/GADD153 via the endoplasmic reticulum and mitochondrial stress pathways, which is similar to other endoplasmic reticulum stress viruses; however, ASFV infection has failed to activate CHOP/GADD153 and inhibit its induction through other stimuli [184]. ASFV induces apoptosis via the mitochondrial pathway but not the receptor-mediated pathway [182]. ASFV can also initiate endoplasmic reticulum stress by activating the ATF6 pathway; however, ASFV blocks the transcriptional induction of specific ATF6 target genes, such as XBP1 and BIP, but not that of calnexin or calreticulin. The protein encoded by the A179L gene of ASFV, a putative Bcl-2 like protein, physically binds to all core death-inducing mammalian Bcl-2 proteins to inhibit apoptosis [185]. A179L protein is expressed in the form of an 18 k Da protein in the early and late stages of macrophage infection [186]. It localizes in mitochondria and endoplasmic reticulum, and it can inhibit the apoptosis of different cell systems. For example, it can inhibit the apoptosis of HeLa cells and insect cells [180]. The A224L gene encodes a member of the inhibitor of apoptosis family, the IAP protein, which inhibits the activation of cysteine proteases and promotes cell survival. Viral IAP not only blocks caspase 3 activation, but also activates NF-κB [187]. Interestingly, ASFV also encodes another I kB-like molecule (i.e., A238L), which interferes with NF-κ B activation [173]. Exogenously expressed ASFV DP71L protein inhibits the activation of the pro-apoptotic protein CHOP [176]. The EP153R gene inhibits apoptosis in viral infection and heterologous expression [188]. The DP96R of the Chinese ASFV 2018/1 strain subverts type I IFN production in the c GAS sensing pathway. DP96R inhibits c GAS/STING and TBK1, but not IRF3-5D-mediated activation of IFN-β, and it inhibits the promoters of the interferon-stimulated response element (ISRE). Furthermore, DP96R selectively blocks activation induced by c GAS/STING, TBK1, and NF-κB promoters, but not by overexpressing p65 [167]. In the pMGFs family, pMGF505-2R, pMGF505-7R, and pMGF505-9R exerted a strong inhibitory effect on IL-1β maturation, especially pMGF505-7R. pMGF505-7R binds to IKK complex to inhibit TLRs/MyD88-dependent pro-IL-1β transcription. Moreover, pMGF505-7R binds to IRF3 to suppress type I IFN signaling [189]. Li et al. found that MGF-505-7R promoted the expression of the autophagy-related protein ULK1 to inhibit the cGAS-STING signaling pathway [106]. It can be seen that pMGF505-7R ASFV-encoded pMGF505-7R plays an important role in virus infection. ASFV infection leads to changes of transcriptional levels of PRRs in some RLR and TLR signaling pathways, anti-viral and inflammatory factors, as well as pro-apoptotic and anti-apoptotic factors, hinting that ASFV infection may activate the RLR and TLR signaling pathways and be involved in the regulation of host apoptosis in many ways [190]. The specific mechanism of ASFV controlling host transcription and the development of the apoptosis process needs to be explored further.

### 6.2. Autophagy

As a highly conserved process, autophagy plays an important role in the occurrence and development of many diseases, including cancer, neurodegenerative diseases, and aging, as well as in both innate and adaptive immunity [191]. Many viruses could promote their own replication by inhibiting the autophagy pathway [192,193]. Some studies have shown that certain viruses could hijack the autophagy pathway to evade the natural immune response and maintain its own replication [194,195,196]. A study has shown that ASFV did not seem to cause LC3 lipidation or autophagosome formation in vero cells [197]. A179L, the viral Bcl-2 homolog of ASFV, is not only an effective inhibitor of apoptosis [198], but also possesses potent autophagy inhibitory activity [197]. A179L could bind with Bcl-2 to hinder the autophagosome formation [197]. Banjara et al. further mutated the A179L binding groove, which could combine with the Beclin BH3 peptide, and found that after mutation, the ability of A179L to bind to Beclin and its ability to inhibit the formation of autophagosomes during starvation were decreased [185]. In addition, ASFV could inhibit autophagy pathway by activated mTORC1 [199]. Interestingly, a complete autophagy process could be induced in vero cells and human embryonic kidney-293T (HEK-293T) cells when transiently transfected with recombinant plasmid-expressing ASFV-E199L protein [200]. E199L protein decreased the expression of PYCR2, resulting in autophagy activation [200]. The function of ASFV proteins has not been fully clarified and the research between autophagy and ASFV infection is still a gap, which awaits further investigation.

## 7. Virus Inactivation

ASFV in the environment and raw pork products is highly stable at low temperatures, which is conducive to survival in cold, humid, and organic environments. Therefore, ASFV can remain infectious for 15 weeks in frozen meat, for 6 months in cured ham, and for 399 days in Parma ham [201]. In liquid fertilizer, stability was observed for more than 100 days. In liquid blood, the virus can survive for 18 months at room temperature, for 6 years at 4 °C, and longer when frozen [202]. The survival of pork products depend on the nature of the reagent, especially the stability/instability to time, temperature, and pH, as well as the inherent properties of the product, such as pH, water activity, processing and storage temperatures, and salinity [203]. ASFV can survive for 18 days in salami, 60 days in pork belly, and 83 days in tenderloin [204]. A recent study in Lithuania exhumed buried carcasses of ASFV-infected wild boars at different time points and locations and re-tested these for the presence of infectious ASFV through in vitro testing and quantitative polymerase chain reaction. Surprisingly, only the viral genome was found, and all virus isolation attempts remained negative [205]. The relationship between the ASFV infection isolated from the cadaver sample, the sensitivity of the animal, and the dose required for oral vaccination needs further study. One study examined virus stability by adding blood from pigs that had been infected with ASFV to different soil substrates. The results showed that soil pH, structure, and environmental temperature play an important role for the stability of infectious ASFV. The virus can be separated from soil within a few weeks, and the virus can also be separated from soil in swamp areas within a few days. ASFV could not be separated from strongly acidic soil [206]. Another study showed that ASFV in pig manure was inactivated after 4 h at 40 °C. Urine ASFV remains infectious for 4 days at 37 °C, and ASFV in feces remains infectious for 3 days at 37 °C [201]. When feeding flies in vitro with blood from ASFV-infected pigs, ASFV-DNA can be detected in the flies’ mouth for at least the following 12 h and can be identified in the sample for 3 days. Infectious virus was detected in fly samples prepared 3 h and 12 h after feeding of the flies. ASFV can survive in leeches for 60–80 days. Adding serum to a serum-free medium can increase the resistance of the virus. For example, at pH 13.4, resistance can last 21 h in serum-free medium and 7 days in medium with serum [207,208].

To date, neither medications nor effective vaccines exist for ASFV. At present, farms mainly control the occurrence of ASFV by improving their applied biosafety standards. ASFV can be inactivated at 60 °C for 30 min [209]. Sodium hypochlorite is one of the compounds recommended for the inactivation of ASFV, as low concentrations have been found to be effective for disinfection [210]. However, its effectiveness has been reduced by long-term storage, and it may be necessary to check its activity before use. To achieve a satisfactory disinfection effect, 0.5% active chlorine concentration is necessary [211]. It was found that calcium hydroxide inactivated ASFV at 1% and 0.5% within 30 min at 4 °C and 22 °C, respectively; sodium hydroxide was effective at 1%, 0.5%, and 0.2% at 22 °C, as well as at 1% and 0.5% at 4 °C. However, at 4 °C, 0.2% sodium hydroxide was ineffective. A 2%, caustic soda solution is the strongest compound that can be used to deactivate ASFV. In case of an ASF outbreak, large surfaces and transport tools can be cleaned with a 2% caustic soda solution and a 1% solution can be used for hand disinfection [212]. Glutaraldehyde can be used as a virucidal agent to also inactivate ASFV, because it disrupts biofilms through protein denaturation and interferes with the metabolism of protein-DNA crosslinks. Cresol was another important disinfectant, which was shown to be 10 times more active than phenol and can thus also be used as an ASFV disinfectant [212]. Tests have confirmed that quaternary ammonium compounds (QACs) at 0.003% were very effective against four enveloped viruses, including ASFV [213]. All of the compounds mentioned above can be used for ASFV disinfection. In addition, heat treatment of ASFV-contaminated infectious blood for the fertilization of field crops has an inactivating effect, but contaminated feed should not continue to be fed to animals [214]. Ozonated water can efficiently inactivate the ASFV [215]. ASFV can be inactivated by pH <3.9 or >11.5 in serum-free medium. The OIE has published methods for the inactivation of ASFV by chemicals or disinfectants: 8/1000 sodium hydroxide (30 min), 2.3% hypochlorite (3 min), 3/1000 formalin (30 min), and 3% n-phenyl phenol and iodine compound (30 min) (Table 6 and Table 7) [216].

It is hard to find the perfect disinfectant against ASFV because there is no global collation effort to gather explicitly described and detailed data regarding disinfectants. Studies have shown that the chemical compounds effective in inactivation of ASFV are as follows [213]:(1)1% formaldehyde;(2)Sodium hypochlorite (0.03% to 0.0075%);(3)2% caustic soda solution (the strongest virucidal agent);(4)Glutaraldehyde, formic;(5)1% sodium or calcium hydroxide (effective at virus inactivation in slurry at 4°C);(6)v Phenols—lysol, lysephoform, and creolin;(7)Chemical compounds based on lipid solvents [212];(8)Multi-constituent compounds such as Virkon (1:100), Lysoformin, and Desoform, Octyldodeceth-20 (OD-20) surfactants, active substances, organic acids, and glycosal;(9)Triton X-100, and NP-40 [217].

Due to the high genotypic diversity and variability, as well as the environmental adaptability of ASFV, prevention and control of this disease seem to be difficult. With the lack of effective vaccines and treatment, high standards of animal farm management with strict biosafety measures are still necessary and effective. It is necessary to further strengthen the understanding of the specific transmission mechanism of ASFV through research and provide ideas for the design of prevention and control measures. The propagation of knowledge regarding ASFV should be enhanced to increase the vigilance of pig farmers, strengthen the management of pig farms and the surrounding environment, and carry out disinfection work to prevent the spread of the ASFV.

## 8. Virus Typing

Over the past decades, in response to the onset of ASFV, ASFV strains have been typed mainly by sequencing the genes p54, p72, and pB602L [218]. The ASFV major capsid protein (p72) gene (B646L) was one of the first genetic targets used to assess the genetic diversity of ASFV on multiple scales [219]. The region most commonly used for genotype designation was the p72-coding region [220]. The ASFV strains prevalent in Europe (except for Sardinia) and Asia belong to the p72 genotype II (Figure 6) [206]. In addition, the genotype was found not to be related to virulence or pathogenicity [98]. Standard ASFV genotypic markers have been established based on partial B646L (p72) gene sequencing. The B646L genotypic marker enabled relatively rapid and simple typing of ASFV strains; however, B646L genotyping analysis did not always yield a sufficient ability to type or distinguish between strains of different virulence [220]. This has made the use of sequencing less effective for the molecular tracking of virus strains in outbreak areas, such as for molecular epidemiology of epidemic conditions. Further analysis of small genomic regions has enabled the identification of different genetic variants in closely related ASFV genotype II isolates. The genotypic resolution could be improved via the combined assessment of the p54 (E183L), p30 (CP205L), and B602L genes [221]. A large-scale molecular epidemiological study using African ASFV isolates identified the presence of a large number of ASFV variants on the African continent [10]. Only the whole genome may be able to achieve higher-resolution discrimination and infer virulence genes. New strain variants with deletion genes can be identified by whole-genome sequencing. However, the correlation between the currently identified ASFV genotypes and virulence remains unclear [89]. To date, whole-genome sequences have been reported for 17 genotype II strains, including the MK333180-Pig/Heilongjiang/2018 (Chinese pig/HLJ/18). Comparison of all genotype II isolates showed that all these genomes were almost identical, showing more than 99.9% identity. A deep sequencing workflow has now been established for reliable and high-quality whole-genome sequences. This workflow is based on target enrichment and uses different sequencing platforms to circumvent specific drawbacks [222].

## 9. Disease Dissemination and Control

### 9.1. Dissemination

ASF is a devastating disease for the pig farming industry with a high mortality in pigs, thus threatening both global pork production and food security. ASF was first reported in East Africa in the early 1900s, and the source of infection was identified as a virus that emerged from an ancient forest [2]. In the following period, ASFV spread to most sub-Saharan African countries [224]. ASFV then spread from Portugal to other countries in Europe, the Caribbean, and Brazil, and was introduced to Georgia in the Caucasus in 2007. From here, it spread to the Russian Federation, Ukraine, and Belarus, and in 2014 to the European Union, Baltic States, and Poland [225]. By 2018, the infection had also spread to Belgium, Hungary, the Czech Republic, Romania, Bulgaria, Slovakia, and Serbia [10]. Finally, in 2018, ASF was also introduced to China, a major pig farming country where half of the world’s pigs are farmed. Within a few months of the first ASF outbreak, ASFV quickly swept through most of China’s provinces. A total of 178 ASF outbreaks (including four wild boar outbreaks) have been reported in 31 provinces [226], cities, and autonomous regions. After its spread in China, ASFV has spread to Mongolia, Vietnam, Cambodia, North Korea, Myanmar, Laos, and the Philippines (Figure 7) [227]. Large-scale epidemics can lead to dramatic reductions in pig populations and increases in the price of pigs and pork products.

As world trade flows intensify, trade in pork products, in particular, plays a key role in sustaining viruses locally, within countries and regions, and in their long-distance transmission [229]. The potential for the spread of ASF through pig trade networks between the member states of the European Union has also been recognized, particularly in cases where a long period of time passes between infection and the reporting of the disease [230]. In China, the inter-regional transport of live pigs and pork products is an important factor contributing to the rapid spread of ASF throughout the country [225]. The feeding of unsterilized food residues is also an important factor in the rapid spread of ASF in China. Contaminants such as clothing, transport trucks, or feed can be a source of infection.

### 9.2. Control

#### 9.2.1. Government

The Chinese government attaches great importance to the prevention and control of ASF. The Chinese authorities in charge of animal health quickly launched a high-priority emergency response plan against this major epizootic. An extensive surveillance network covering all the provinces has thus been put in place and various support measures have been applied, in particular compensation for breeders whose animals had been slaughtered and a credit scheme for affected producers. In addition, a joint coordination body for ASF prevention and control has been created under the leadership of the Ministry of Agriculture and Rural Affairs, with the participation of twenty other ministries including those in charge of transport, customs, and market surveillance. All levels of local government have assumed territorial management responsibilities and put in place a support policy to encourage all actors to participate in the prevention and control of ASF. In addition, the Chinese government has developed a strategy integrating both legal and scientific aspects of the fight against ASF through a comprehensive set of measures covering the entire pork chain “from farm to fork”. The plan provided for daily reports to be issued by the ASF monitoring network and various effective measures including blockade and slaughter orders, restriction of pig movements and control of swill intended for animal feed, in order to block the transmission routes of the virus, as far as possible [231].

The control of ASF is a very important task, which is related to the food security and economic stability of a country. Currently, there is no safe, effective, and commercially available vaccine for ASF. China’s control efforts of ASF focus on the elimination of contaminants and the blocking of transmission routes, which will remain a long-term effort [225]. ASF cannot be controlled by farmers alone but requires the involvement of all parties involved in the pork food industry. An epidemiological study that investigated 68 outbreaks in China from August to November 2018 showed that 19% were caused by irregular transport of live pigs and pork products, 46% by vehicles and people carrying the virus, and 34% by slop feeding [226]. The development of policies by animal health authorities to guide practitioners toward the implementation of effective measures can reduce the occurrence of ASF outbreaks. For example, regional biosafety regulations should be formulated, cross-regional trade of live pigs should be stopped, and the sale, burial, and littering of diseased animals should be prohibited. Efforts should be made to establish a binding legal framework for control programs in countries with current ASF endemics. The most vital components of control measures are timely and reliable diagnosis, eradication of infected herds, establishment of restricted areas, movement restrictions, and tracing of potential contacts. Taking China as example, the Ministry of Agriculture has strengthened both the supervision and management of the movement of pigs and their products and has implemented measures such as the registration of pig transport vehicles, inspections at the transport stage, and testing at the slaughter stage. The control of the movement of pigs and the transport of products appeared to be effective, thus leading to a decrease in the proportion of outbreaks from an original 35% to 15% [232]. The Ministry of Agriculture and Rural Affairs of the People’s Republic of China issued a ban on the feeding of food waste to pigs, which decreased the proportion of outbreaks caused by the feeding of food waste from an initial 50% to 44% [233].

Grassroots farmers generally have very low awareness of virus control measures, and veterinarians play a very important role in managing ASF. A total of 54% of outbreaks reported in China in 2018 originated from small- and medium-sized farms, further illustrating the lack of biosecurity awareness among decentralized family farms [234]. In China, many large farms do not employ a veterinarian, and in specific areas, only one veterinarian is available for several farms. The establishment of veterinary teams often forces practitioners to be more observant of control and prevention policies. The control of ASF is still mainly preventive, and China currently invests more human and material resources in preventing the spread of ASF. The government initiated a financial compensation policy to compensate farmers for the slaughter of sick animals. Without effective financial compensation, farmers would sell all their sick pigs quickly to reduce their financial losses, resulting in the circulation of sick pigs to the market and an increased risk of disease transmission, which is not conducive to the prevention and control of ASF. The government also provides reliable testing services or testing methods for farmers, which increases early detection. Such early detection allows for timely detection of sick pigs and enables the application of stronger measures to prevent the spread of the disease. To date, many national authorities have issued policies and documents for the prevention and control of ASF.

#### 9.2.2. Breeding Farms

Farmers are the main force for controlling ASF and shoulder very important responsibilities. While controlling ASF, they are also protecting the safety of their own economic assets, and to this end, it is desirable for farms to keep ASFV out of their environment. Therefore, it is vital to build a sound biosecurity system, which includes both on-farm and off-farm biosecurity systems. For this, it is recommended that farms are bounded by a fence, thus establishing both an on-farm area and an off-farm area. Pig farms without fencing or with inadequate fencing are advised to upgrade their fencing system. The fence could stop the entry of outside wild animals, as these may induce the risk of spreading ASFV. The construction of a farm biosecurity system includes the following main steps:(1)Keep materials that will eventually enter the piggery out of the piggery at first, until they have been disinfected in the external disinfection area.(2)The exit for the selling of pigs should be built on the outside of the enclosure.(3)No visiting vehicles should be allowed inside the enclosure.(4)Staff should be strictly bathed, and their clothing disinfected before entering the farm.(5)Rodent-proofing should be established along the fence to keep wild animals out.(6)Pens should be regularly disinfected.(7)The breeding density should not be too high.(8)Pigs should be regularly tested.(9)Emergency treatment should be initiated after an outbreak of illness.

#### 9.2.3. Emergency Handling

After the ASFV has infected its host, the virus can be detected in the blood, urine, and feces of pigs [201]. Environmental contamination of ASFV by infected animals (especially feral pigs) may contribute to the spread of the virus, as the virus has been found to be able to survive in the environment for some time [235]. However, the survival time of infectious ASFV in the environment is shorter than the survival time indicated by viral DNA testing [107]. Contaminated environments (maintained at an average temperature of approximately 21 °C and an average relative humidity of 33–37%) were infectious on day 1 after the removal of infected animals, but similar environments were unable to cause infection after 3 days (or more). This short cycle of virus transmission suggests that feces- or urine-contaminated material may only play a minor role in the long-term transmission of the virus [236].

During daily pig feeding and health observations, if decreased appetite is observed in specific pigs, strict pig group isolation urgently needs to be adopted, and diseased pigs need to be tested on the spot as soon as possible. Once the diagnosis of ASF is confirmed, affected pigs should be immediately disposed of without harm, while the pig sheds and the areas through which sick pigs had passed need to be strictly disinfected. At the same time, the awareness of surrounding herds needs to be raised to ensure that other pigs are not infected. This series of operations, including the precise detection, diagnosis, removal of the virus, and disinfection are referred to as “precision culling”. This needs to be a rapid response, which should be accompanied by decisive treatment and thorough disinfection.

## 10. Vaccination

ASFV is a virus with a large genome and research into protective antigens and virulence genes is an ongoing process [237]. ASFV encodes a range of immune escape proteins that create favorable conditions for self-proliferation by regulating host cell protein expression. ASFV also interferes with the innate immune system to suppress and evade host immune responses. So far, preventive vaccinations and other treatments are still not available, and inactivated ASFV vaccines have been strictly prohibited in the European Union and other countries [206]. Studies have shown that vaccination with inactivated ASFV vaccines is essentially a non-protective strategy [238]. Although previous studies have shown that protective immunity can be achieved with attenuated vaccines, virulence, immunogenicity, and more importantly, viral phenotype and antigen diversity issues, as well as the lack of strain cross-protective immunity, continue to affect ASF live attenuated vaccines (Table 8, Table 9 and Table 10) [239]. Many studies have assessed gene-deficient vaccines in the hope of deleting virulence genes and interferon suppressor genes for several strains of ASF (Table 11) [237]. Deletions were either made individually or in combination, and the deleted ASF virulence-associated genes included 9GL (B119L), UK (DP96R), CD2v (EP402R), DP148R, and different members of the multigene family (MGF) [240]. The gene knockout strain obtained after deletion of the gene is capable of producing protective effects in immunized animals [142]. Gladue et al., constructed a recombinant virus lacking the A137R gene (ASFV-G-ΔA137R) and confirmed that ASFV-G-ΔA137R exhibited a completely attenuated phenotype, so it has become a novel potential live attenuated vaccine candidate to protect the animals from the epidemiologically relevant ASFV Georgia isolate [152]. However, in genotype VIII strains, deletion of the same genes does not result in attenuation of virulence, and the genetic background of ASFV affects the phenotype of deletion mutants [241]. Chen et al., proposed a live attenuated vaccine candidate strain with seven gene deletions, encoding the genes MGF505-1R, MGF505-2R, MGF505-3R, MGF360-12L, MGF360-13L, MGF360-14L, and CD2v (EP402R). The results showed that the duration of protection was highly dependent on the dose of immunization. Although high doses can provide long-term protection, 80 days after the last vaccination, double vaccination with medium doses could no longer ensure complete protection [242].

In recent years, a number of research studies have evaluated genetically engineered vaccines. Several ASFV proteins (p30, p54, p72, CD2v, EP153R, p12, D117L, and pp62) have been studied and reported as major targets for ASF vaccines. The immunoprotected capacity of these proteins has been tested, including individual ASFV antigen targets or multi-target cocktails for vaccination [225]. In currently available subunit vaccines, most of the utilized ASFV protective antigens are not sufficient to provide complete protection, and subunit vaccines offer safety advantages over attenuated vaccines. Previous studies have shown that immunization of domestic pigs with a mixture of replication-deficient adenoviruses expressing ASFV antigens p32, p54, pp62, p72 or A151R, B119L, B602L, EP402RPRR, B438L, K205R-A104R, pp62, p72, and pp220 elicited potent antigen-specific antibodies, IFN-γ cells, and cytotoxic T lymphocyte responses [201].

Recently, researchers have developed a cocktail vaccine containing 35 adenovirus-vectored ASFV antigens and tested it in wild boar to assess its protective effect against the virulent ASFV Arm07 isolate. In their study, the cocktail contained adenovirus expression of the above ASFV antigens and other viruses, such as p220, p72, p15, B602L, p62, p32, p54, EP153R, p10, K205R, A104R, EP402R PRR, A151R, B119G, K196R, CP80R, B438L, R298L, NP419L, K145R, B385R, F334L, CP312R, H108R, F16 5R, F778R, S273R, MGF100-1L, B66L, NP868R, H339R, I329L, A224L, MGF505-6R, and B175L [240]. Ultimately, a mixture of 35 adenovirus-vectored ASFV antigens was tested for its protective efficacy in wild boars. No specific maintenance antibodies against ASFV were detected in any of the treated animals by enzyme-linked immunosorbent assay (ELISA) before challenge [240]. This result is inconsistent with the strong response of primary and memory antibodies observed in domestic pigs inoculated with four or seven adenovirus-vectored ASFV antigens [240]. Many countries and institutions are committed to the development of an ASF vaccine. It is very difficult to eradicate ASF in China (and other countries), and therefore vaccine development is necessary, requiring the support of ASFV virology and knowledge of relevant genomics. However, the safety of the vaccine must be guaranteed, and experimental animal models should exist to evaluate its safety. Many reviews summarize the methods used to study the ASF vaccine and its protective effect [225,237,243,244,245]. At this stage, limited understanding of protective immunogens and virulence-related genes of ASFV has impeded the development of ASFV vaccines. It is essential to find novel and highly efficient ASFV epitopes that could stimulate stronger immune responses in the prevention of ASFV infection. As is known to all, live attenuated vaccine can confer a great protection on pigs and is currently the most feasible approach to serve as an effective ASF vaccine [246]. The safety of live attenuated vaccine can be improved by deleting multiple virulence-related genes. Due to its ideal protective effect, live attenuated vaccines are still the research development trend for ASF vaccines in the short term. However, before the actual application, it is necessary to conduct a comprehensive evaluation of the safety risks including virulence re-enhancement, adverse reactions, and persistent infection of the candidate strains [225]. Significantly, the development of ASFV vaccine still relies on primary swine macrophages, which has led to an increase in the time and cost of ASFV vaccine development. There is an urgent need to find a suitable and low-cost cell line, which is of great significance for the development of an ASFV vaccine. ASF subunit vaccines, DNA vaccines, and viral vector vaccines have great research potential with high security. However, there is currently no vaccine that can provide complete immune protection against the deadly ASFV strain. So far, no specific viral protein has a sufficient ability to trigger complete immune protection. It is necessary to carry out in-depth research on ASFV structure analysis, identification of antigen gene or protein, immune protection mechanism, and at the same time develop an efficient antigen expression system to improve the genetic stability of vaccine strains that can better induce the production of high-level antibodies in the body.

Experimental live attenuated vaccine viruses, generated by deleting virus genes associated with virulence, have been shown to be effective [13,14,15,16,17,18]. One of these vaccines, ASFV-G-∆I177L [14], induces effective protection against its parental virulent virus strain, Georgia 2007, as well as a virulent Vietnamese field strain isolated in 2019 [19]. This vaccine has been selected for further development and commercialization, which includes a thorough assessment of its safety characteristics. Tran evaluated the safety profile of an efficacious live attenuated vaccine candidate, ASFV-G-∆I177L. Results from safety studies showed that ASFV-G-∆I177L remains genetically stable and phenotypically attenuated during a five-passage reversion to virulence study in domestic swine. In addition, large-scale experiments to detect virus shedding and transmission confirmed that even under varying conditions, ASFV-G-∆I177L is a safe live attenuated vaccine [20].
life-12-01255-t008_Table 8Table 8Antigen-based African swine fever virus (ASFV) vaccines.Vaccine TypeASFV Target Protein (Strain)Number of Immunizations; Dose, AdjuvantSpecific/Neutralizing AntibodiesT Cell ResponseChallenge Strain; DoseClinical OutcomeReferenceBaculovirus-expressed proteinsCD2v (E75CV)3×; 0.5–1 × 10^7^ HAU + Freund’s adjuvantYes; NoNAE75; 4 × 10^2^100% protection, *n* = 3/3[247]Baculovirus-expressed proteinsp30, p54, p54 + p30 (E75)3×; 100 μg + Freund’s adjuvantYes; YesNAE75; 5 × 10^2^50% protection, *n* = 3/6[248]Baculovirus-expressed proteinsp54/p30 chimera (E75)5×; 100 μg + Freund’s adjuvantYes; YesNAE75; 5 × 10^2^100% protection, *n* = 2/2[249]Baculovirus-expressed proteinsp54 + p30 + p72 + p22 (Pr4)4×; 200 μg + Freund’s adjuvantYes; YesNAPr4; 10^4^Slight delay of clinical disease and viremia; no protection, (*n* = 0/6)[250]HEK cell-expressed proteinsp72, p54, p12 (Georgia 2007/1)2×; 200 μg/antigen + TS6 adjuvantYes; NASomeNANA[251]DNA (pCMV)SLA-II/p54/p30 fusion (E75)3×; 600 μgYes; NoYesE75; 10^4^No protection, (*n* = 0/4)[252]DNA (pCMV)sHA/p54/p30 fusion (E75)3× and 4×; 600 μgYes; NoYesE75; 10^4^No protection, (*n* = 0/6)[253]DNA (pCMV)Ub/sHA/p54/p30 fusion (E75)2× and 4×; 600 μgNot detectableYesE75; 10^4^Partial protection, (2 immunizations, *n* = 2/6; 4 immunizations, *n* = 1/6)[253]DNA expression library80 ORFs fragments fused with Ub (Ba71V)2×; 600 μgYes—after challenge; NAYes-after challengeE75; 10^4^60% protection (*n* = 6/10)[254]BacMamsHA/p54/p30 fusion (E75)3×; 10^7^ PFUNo (only after challenge); NoYesE75; 2× sublethal challenge 10^2^Partial protection (*n* = 4/6)[255]Adenovirusp30 + p54 + pp62 + p72 (Georgia 2007/1)2×; 10^10^ or 10^11^ per Ad5-antigen + adjuvantsYes; NAYesNANA[256]AdenovirusA151R +B119L +B602L +EP402RΔPRR +B438L +K205R +A104R (Georgia 2007/1)2×; 10^11^ per Ad5-antigen + adjuvantYes; NAYesNANA[156]Vaccinia virus Ankarap72, C-type Lectin, CD2v (Georgia 2007/1)2×; rVACV-ASFV 10^7^ TCID_50_No; NAYesNANA[251]Alphavirus RPsp30, p54, p72, sHA/72 (Ba71V)3×; 2–4.5 × 10^7^ RPsYes; NANANANA[257]DNA–ProteinCombinations of DNA and protein: p15, p30, p35, p54, p72, CD2v, CP312R, g5R (Georgia 2007/1; Ba71V)3×; 100 μg per DNA, 100 μg protein + ISA25 adjuvantYes; YesSomeNANA[258]DNA–ProteinProteins: p15, p35, p54, p17; DNA: CD2v, p72, p54, p30, p17 (Georgia 2007/1; Ba71V)3×; 100 μg per DNA, 100 μg protein + ISA25 adjuvantYes; NoSomeArmenia 2007; 360 HAUNo protection; disease enhancement[259]DNA prime + vaccinia virus boost47 antigens (Georgia 2007/1)Prime 2×: 10 μg pCMV-DNA + CpG oligo adjuvant;Boost 2×: 10^8^ PFU rVACV-ASFVYes; NoYesGeorgia 2007/1; 10^4^No protection; reduced viral load, higher clinical scores[260]Vaccinia virus prime + protein boostp72, C-type Lectin, CD2v (Georgia 2007/1)Prime: rVACV-ASFV 10^7^ TCID_50_; Boost: 200 μg/antigen + TS6 adjuvantNAYesNANA[251]Alphavirus RP prime + live attenuated ASFV boostp30 (Ba71V) + OURT88/3Prime 2×: 2–4.5 × 10^7^ RPs; Boost: 10^4^ TCID_50_ OURT88/3Yes; YesNANANA[257]p30 referred to as p32; CD2v also referred to as HA = hemagglutinin; NA = not available.





life-12-01255-t009_Table 9Table 9Naturally attenuated strains.StrainVirulenceChallengeProtectionReferencesNH/P68LowHeterologous strain L60100%[261]OUR T88/3LowHeterologous strain Benin 97/185.70%[262]OUR T88/3LowHomologous strain OURT88/150–100%[263]Lv17/WB/Rie1LowHomologous strain HAD Latvian ASFV100%[264]
life-12-01255-t010_Table 10Table 10Recombinant-attenuated strains.StrainVirulenceDeleted GenesChallengeProtectionReferencesOUR T88/3LowDP71L and DP96RHomologous strain OURT88/1100%[261]Georgia 2007/1HighDP96R(UK) and B119L(9GL)Homologous strain Georgia 2007/1100%[18]Benin 97/1HighDP148RHomologous strain Benin 97/1100%[164]Georgia 2007/1HighMGF505/360(6)Homologous strain Georgia 2007/1100%[97]Georgia 2007/1HighMGF505/360 and B119L(9GL)Homologous strain Georgia 2007/1100%[265]Benin 97/1HighMGF505/530/360Homologous strain Benin 97/1100%[266]BA71HighEP402R(CD2v)Heterologous strain Georgia 2007/1100%[13]Georgia 2010HighEP402R(CD2v)Homologous strain Georgia 2010100%[16]HLJ/2018HighMGF505/360(6) and EP402R(CD2v)Homologous strain HLJ/2018100%[242]Georgia 2010HighI177LHomologous strain Georgia 2010100%[14]Georgia 2007/1HighL83LDo not verifyDo not verify[267]Georgia 2007/1HighB119L, DP71L and DP96RHomologous strain Georgia 2007/10[268]NH/P68LowA276RHeterologous strain virulent Arm070[269]
life-12-01255-t011_Table 11Table 11LAV candidates.GenesStrainsGenotypeMinimal Protective DoseRouteChallengeGene FunctionReferencesA137RGeorgia2007/1II102HAD50IMGeorgia2007/1Unknown[152]I226RSY18II102HAD50IMGeorgia2007/1Unknown[270]L7L-L11LSY18II103HAD50IMSY18Unknown[271]MGF505/360and EP402RHLJ/18II103HAD50105HAD50IMONHLJ/18Hemadsorbing andinhibition of type I[242]EP402RBa71VI104HAD50IMBa71VE75Georgia2007/1Interferon responsesHemadsorbing[13]I177LGeorgia2007/1II102HAD50106HAD50IMONGeorgia2007/1Georgia2007/1Unknown[15,16]


## 11. Diagnosis

Stringent quarantine and testing require rapid and accurate laboratory diagnosis, which needs to provide significant data for epidemiology, enable the early detection of the disease, and reducing the spread of the ASFV. The optimum diagnosis of ASF should have a consistently high sensitivity and specificity and be designed for easy handling and high-throughput application.

PCR is considered the standard for early diagnosis of ASF because of its superior sensitivity, specificity, and high throughput capability for the detection of the ASFV genome from clinical samples taken from domestic porcine, wild swine, and ticks [272,273]. To date, a variety of PCR assays, including conventional and real-time (qPCR), have been developed and validated using the highly conserved p72 protein gene encoding ASF. With regard to other molecular assays, isothermal amplification analysis may be a more inexpensive diagnostic substitute for PCR and can be used in field conditions. However, these detection methods are not recommended for recovered or virus-carrying animals, since the genomic detection level is significantly below that of the PCR. Mur et al. confirmed the presence of ASFV antibodies in porcine oral fluid samples of all animals from the early post-infection period to the end of the test, using ELISA and international protocol test [274]. Blood serum adsorption tests, virus isolation, fluorescent antibody test antigen detection, ELISA, conventional PCR, and real-time PCR are the most widely used methods for the detection of ASFV [275]. Blood adsorption and virus isolation are sensitive and reliable confirmatory methods for the detection of infectious viruses, but they are also laborious and not suitable for quick routine diagnosis. In addition, virus isolation is not susceptible to lower levels of viremia in animals and is compromised in the presence of antibodies [276]. Serological antibody testing should be performed in parallel with antigen testing to enhance the sensitivity and specificity of the obtained results [6]. In recent years, many methods and kits for the detection of ASFV have been released in almost every country with ASF outbreaks, and the number of commercially available kits for ASFV genomic testing has increased notably.

Recently, Schroder et al. evaluated the sensitivity and specificity of seven commercially available ASFV real-time PCR detection kits and three Taq polymerases on 300 well-characterized wild boar samples collected in Belgium during the 2018–2019 ASF outbreak. The study confirmed that all commercial kits and two Taq polymerases are suitable for ASFV testing by diagnostic laboratories. The most prominent commercial kits include the following: Virotype ASFV 2.0 PCR kit (Indical, Leipzig, Germany); Adiavet ASFV Fast Time (Adiagen, Ploufragan, France); Bio-T kit ASFV (Biosellal, Dardilly, France); VetMax ASFV Detection kit (Thermofisher, Lissieu, France); RealPCR ASFV DNA Test (IDEXX, Hoofddorp, The Netherlands); VetAlert ASF PCR Test Kit (Tetracore, Rockville, MD, USA); and IDGene™ African Swine Fever Duplex (ID.vet, Grabels, France) [277].

Adsorption of red blood cells (hem adsorption) on the surface of macrophages infected with African swine fever virus (ASFV) is a unique phenomenon allowing determination of virus infectious titer in hem adsorption unit (HAU) and differentiation of virus strains phenotypically. In the meantime, hem adsorption of a particular ASFV strain can by inhibited by homologous anti-ASFV serum containing antibody to the serogroup-specific virus protein (CD2v). Here, we describe a hem adsorption inhibition assay (HADIA) to phenotype ASFV strains for one of the nine known serogroups using blood-derived swine macrophages [278]. HADIA is a powerful method in ASFV immunopathology and vaccine research since it provides additional antigenic and phenotypic characteristics of virus strains that cannot be defined by other assays. HADIA-based ASFV classification was developed to differentiate virus antigenic types (serogroup). Nine ASFV serogroups (SG) were defined and subsequently thoroughly characterized at the Federal Research Center for Virology and Microbiology (Pokrov) [279] (Table 12).

## 12. Discussion

Since the first case of ASF occurred in Liaoning Province in northern China in 2018, the ASF epidemic has quickly spread to central, eastern, and western China, and ultimately, to southern provinces, challenging the Chinese breeding industry with unprecedented severity. Over a century of spread and evolution, ASF has threatened the world’s pig herds more than ever. As there is no ASF vaccine, Chinese defense and control of the ASF epidemic largely depends on biosecurity measures. At the same time, practitioners have weak biosecurity awareness, and strict biosecurity has not formed a strong line of resistance to viruses. Although a number of countries banned the ASF vaccine, the development of an ASF vaccine will be beneficial for the control and eradication of ASF based on the current situation in China.

The complex structure of ASFV, its large genome, and the diversity of viral genomes among different strains add to the difficulty of developing a vaccine. The structure and function of the main ASFV proteins, infection, and immune mechanisms must be fully understood, and major immunogens must be identified. Despite the availability of basic information about the ASFV replication mechanism and the process of virus entry/internalization and endosomal transport, many details about the specific process of virus replication still remain unclear. During the entire viral replication cycle, the host gene-encoded enzyme activity and many functions of the viral gene-encoded enzyme remain unknown. To broaden the understanding of the ASFV proteome and the functions of individual proteins, the rational design of targeted vaccine methods is essential. For example, A179L/A224L genes play a role in inhibiting cell apoptosis, but their expression in virus-infected cells and the effect of virus-infected organisms on the infection itself remains to be confirmed. A deeper understanding of the function of ASF protein helps to better understand the pathogenesis of ASF. It is therefore necessary to perform accurate and high-quality whole-genome sequencing of virus genes. In addition, little is known about the mechanism of virulence genes and virus–vector–host interaction.

Scientists have studied various vaccine strategies for ASF and evaluated multiple ASFV-specific targets. However, relevant knowledge of virus receptors, innate immune response, and virus–host interaction still remains limited. So far, the exact nature of the host’s protective response has not been fully determined. The protective antigen remains to be identified, and the ASFV targets that play important roles in virulence and immunopathology or protection still remain unknown. Nevertheless, a number of laboratories are dedicated to the development of ASF vaccines and have made groundbreaking discoveries. At present, subunit vaccines may be a long-term choice for vaccine strategies. Scholars suggested that ASFV strains, attenuated via the knockout of virulence genes, might have a better prospect. However, they have not ruled out that the attenuated vaccine adds the risk of anti-virulence and persistent infection. To develop live attenuated vaccines, relevant safety characteristics and minimum standards of live attenuated vaccines must be formulated, and animal immunity and experimental challenge data should also be gathered. Furthermore, it is important to ensure the effectiveness and safety of vaccines before their promotion and application.

Until a vaccine becomes available, biosecurity may be the most important measure to prevent the spread of ASF. The research on ASFV media should be accelerated, including animal factors, human factors, and environmental factors. Only through understanding the role various factors play in the spread of the virus can more purposeful measures to prevent and control be taken. Similarly, ASF pathological diagnosis and laboratory diagnostic technology research also occupy an important position in the prevention and control of ASF.

China has initiated emergency research on major basic scientific issues related to ASFV, such as vaccine development, diagnostic technology, epidemiology, special disinfectants, and disease vector control. It has been shown that the prevention and control of ASF have a long way to go, and cooperation with the international community is required to overcome prevalent obstacles of vaccine development.

## Figures and Tables

**Figure 1 life-12-01255-f001:**
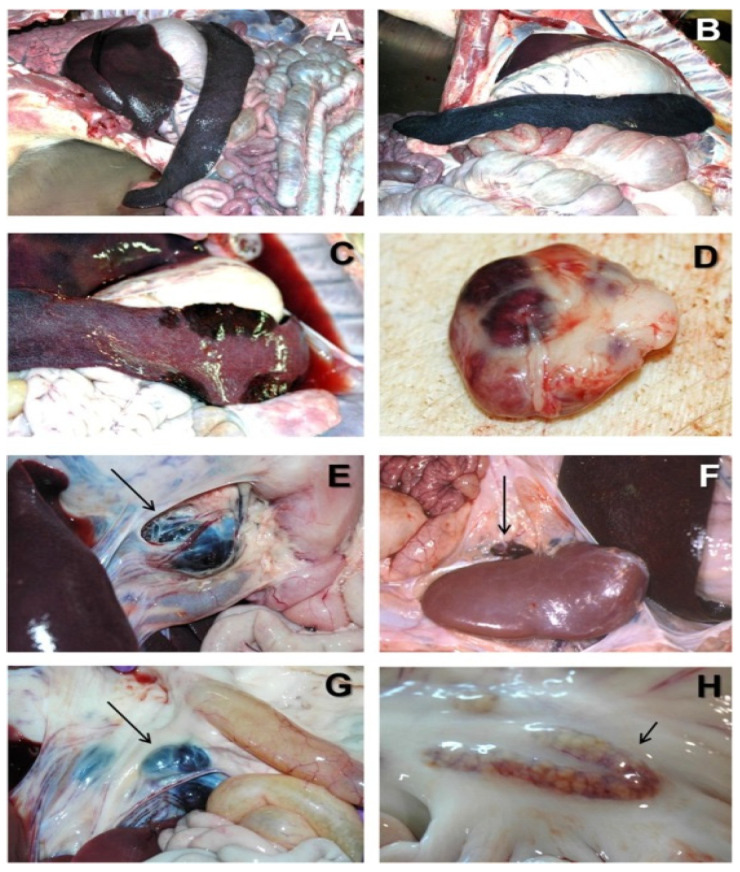
Acute ASF post-mortem examination [7]. (**A**) Severe hemorrhagic splenomegaly observed at the opening of the abdominal cavity of an animal with acute ASF. The liver is severely congested. (**B**) Very large, dark-colored spleen with rounded edges (hemorrhagic splenomegaly), occupying a large volume of the abdominal cavity in acute ASF. (**C**) Multiple areas of partial hemorrhagic splenomegaly in the spleen from an animal with subacute ASF. (**D**) Multifocal hemorrhages in a lymph node with a marbled appearance in acute ASF. (**E**) Severe hemorrhagic lymphadenopathy in the gastrohepatic lymph node (arrow) in acute ASF. (**F**) Severe hemorrhagic lymphadenopathy in the renal lymph node (arrow) in acute ASF. (**G**) Severe hemorrhagic lymphadenopathy in the ileocecal lymph node (arrow) in acute ASF. (**H**) Moderate hemorrhagic lymphadenopathy in the mesenteric lymph node (arrow) in acute ASF.

**Figure 2 life-12-01255-f002:**
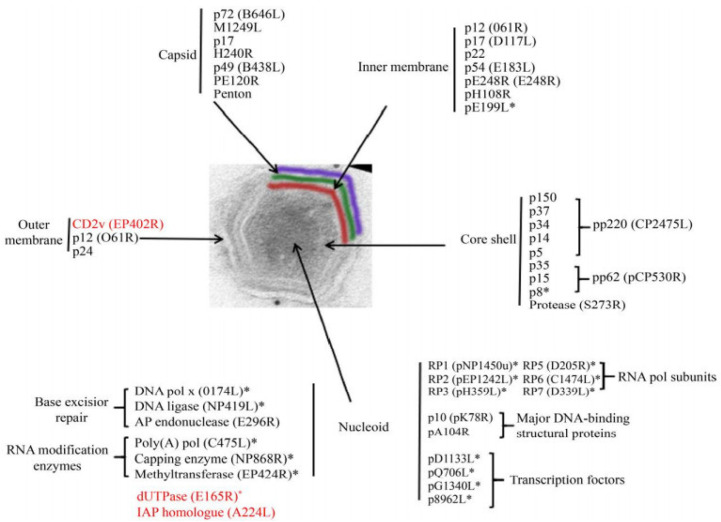
ASFV protein [49]. The distribution of proteins marked with an asterisk (*) was inferred from the predicted or known roles; the genes marked in red are nonessential genes. (adapted with permission from Alejo et al. [49]).

**Figure 3 life-12-01255-f003:**
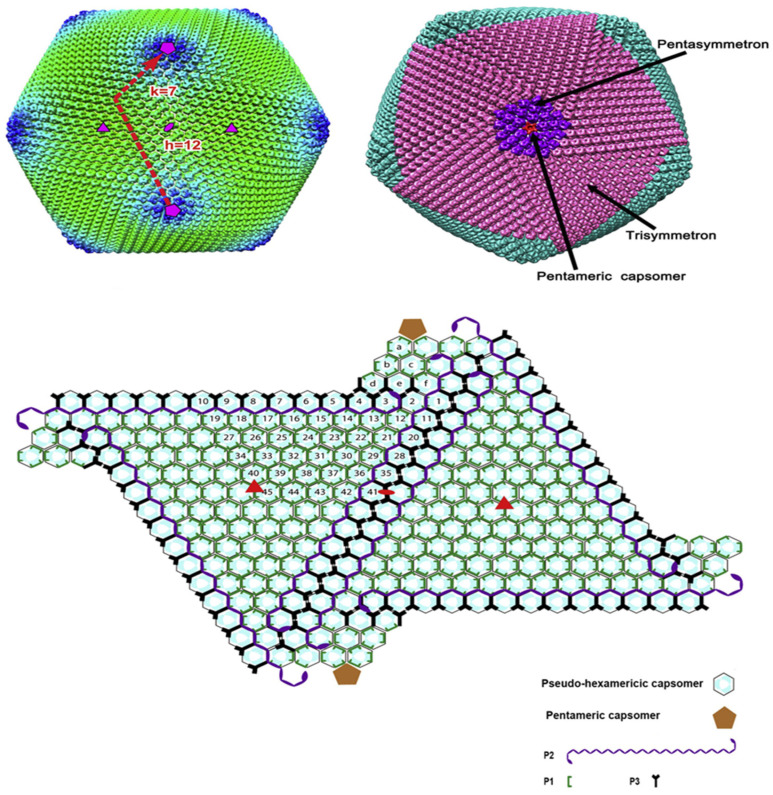
Structure of African swine fever virus (ASFV) capsid icosahedron with average cryo-EM reconstruction. The virus capsid is colored according to the radial distance from the center of the virus. The T numbers, including h and k vectors, are indicated. Three same speed and five same speed are colored in pink and purple, respectively. Graphical organization of both capsomer and small capsid protein are observed from inside of the virus shell. The capsomers of pseudo-hexamers are lining. Each cyan dot in the hexagon represents a p72 subunit. Icosahedron (three times) and the 2-fold axis are shown as a solid red triangle and oval, respectively. Pentose protein and minors’ capsid proteins have different shapes, and as shown in the picture, their colors also differ [55].

**Figure 4 life-12-01255-f004:**
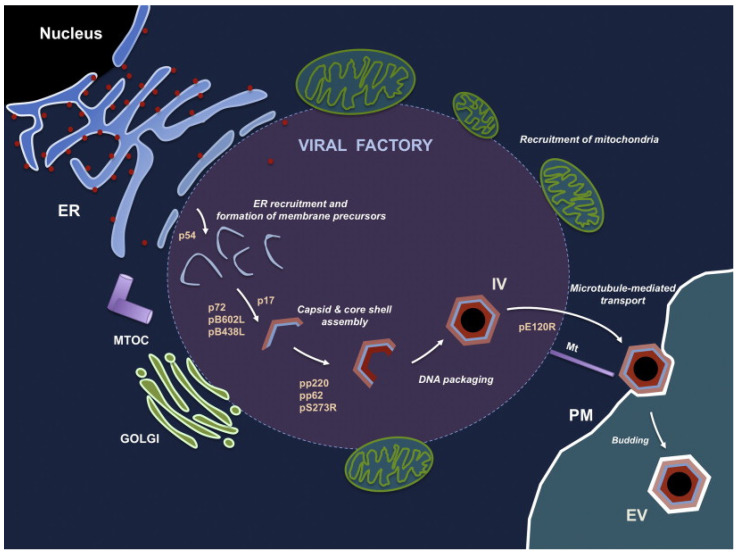
The process of ASFV generation. ASFV morphogenesis occurs in the perinuclear virus factory near the Golgi complex and the microtubule organization center. The first sign of ASFV assembly is the formation of a viral membrane precursor from the endoplasmic reticulum pool. After the capsid is gradually assembled, this viral membrane precursor becomes the polyhedral intermediate. Along with the formation of the capsid, the virus core is assembled under the inner envelope of the virus. The mature virus factory inside the cell is then transferred to the cell surface and leaves the host cell by budding from the plasma membrane. Abbreviations: ER, endoplasmic reticulum; MTOC, microtubule organizing center; PM, plasma membrane; IV, intracellular virus; EV, extracellular virus [47]. (adapted with permission from Andres and Salas [47]).

**Figure 5 life-12-01255-f005:**
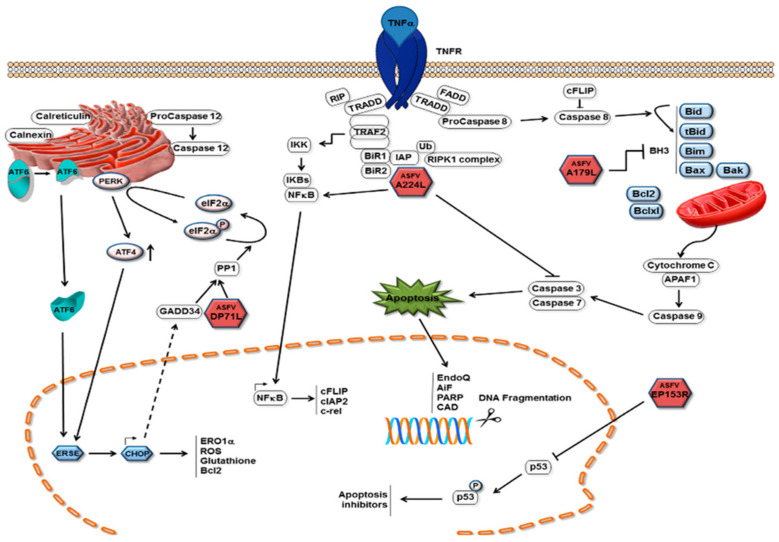
Mechanisms of apoptosis inhibition by ASFV. Pathways with which ASFV inhibits the induction of apoptosis in infected cells are depicted as red icosahedra with the name of the protein presented inside. The ASFV pA179 L Bcl-2 family protein binds to and inhibits several BH3-only domain pro-apoptotic proteins. The pA224 L IAP-family protein binds to (and inhibits) caspase 3 and activates NF-κB signaling, thus increasing the expression of anti-apoptotic genes, including cFLIP, cIAP2, and c-rel [3].

**Figure 6 life-12-01255-f006:**
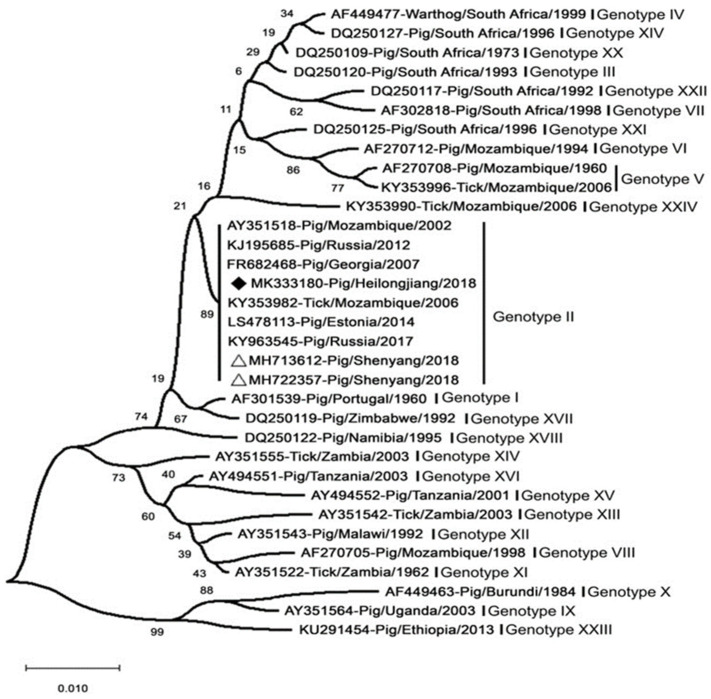
Analysis of the MK333180-Pig/Heilongjiang/2018 (Chinese pig/HLJ/18) based on its partial p72 gene. The white triangles mark the ASFV sequences from the first case in China. Scale bars indicate nucleotide substitutions per site. ASFV isolated in China and Russia share high homology and belong to genotype II [223].

**Figure 7 life-12-01255-f007:**
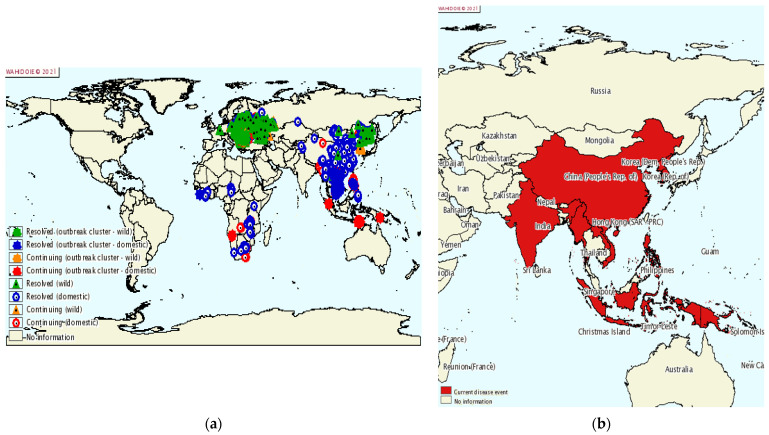
ASF epidemic. (**a**) Global ASF epidemic situation from 2015 to 2020; (**b**) prevalence of ASF in Asia from July to December 2020 [228].

**Table 1 life-12-01255-t001:** Main lesions observed in the different forms of ASF [6].

Symptom	Peracute ASF	Acute ASF	Subacute ASF	Chronic ASF
Fever	High	High	Moderate	Irregular or absent
Thrombocytopenia	Absent	Absent or slight (late)	Transient	Absent
Skin	Erythema	Erythema	Erythema	Necrotic areas
Lymph nodes	–	Gastrohepatic and renal with marbled aspect	The majority of lymph nodes resemble a blood clot	Swollen
Spleen	(-)	Hyperemic splenomegaly	Partial hyperemic splenomegaly or focal infarct	Enlarged with normal color
Kidney	(-)	Petechial hemorrhages, mainly in cortex	Petechial hemorrhages in cortex, medulla and pelvis; perirenal oedema	(-)
Lung	(-)	Severe alveolar oedema	(-)	Pleuritis and pneumonia
Gall bladder	(-)	Petechial hemorrhages	Wall oedema	–
Heart	–	Hemorrhages in epicardium and endocardium	Hemorrhages in epicardium and endocardium; hydropericardium	Fibrinous pericarditis
Tonsils	–	–	–	Necrotic foci
Reproductive alteration	–	–	Abortion	Abortion

**Table 2 life-12-01255-t002:** Composition of white blood cells during acute ASFV infection in swine [31].

Cell Types	The Percent (%) of Cells								ANOVA
	Control	1 dpi	2 dpi	3 dpi	4 dpi	5 dpi	6 dpi	7 dpi	*p*
Lymphoblasts	0	0	0.5 ± 0.1	3.2 ± 0.9 **	3.6 ± 0.9 **	18.9 ± 4.8 **	10.3 ± 2.7 **	2.8 ± 0.7 **	<0.001
Small lymphocytes	34.2 ± 7.6	18.6 ± 4.3 *	25.7 ± 7.0	12.6 ± 3.5 *	11.2 ± 2.9 *	8.5 ± 2.6 *	8.0 ± 2.0 *	3.3 ± 1.0 *	<0.001
Medium lymphocytes	13.2 ± 3.5	10.4 ± 2.2 *	10.6 ± 2.3 *	3.8 ± 0.8 *	7.1 ± 1.8	7.2 ± 2.1	8.4 ± 2.2	3.7 ± 0.8 *	<0.001
Large lymphocytes	8.5 ± 3.1	11.0 ± 2.1	6.4 ± 1.2	6.3 ± 1.9	5.3 ± 1.1	5.4 ± 1.2 *	6.5 ± 1.9	6.5 ± 1.2	<0.001
Reactive lymphocytes	0	6.1 ± 2.0 * *	3.7 ± 0.8 **	3.8 ± 0.9 **	8.0 ± 2.0 **	4.5 ± 1.1 **	4.7 ± 1.1 **	14.8 ± 2.9 **	<0.001
Atypical lymphocytes	0	4.3 ± 1.1 **	8.0 ± 1.8 **	7.0 ± 1.9 **	4.4 ± 1.0 **	5.4 ± 1.2 **	10.3 ± 2.8 **	14.8 ± 3.3 **	<0.001
Monoblasts	0.5 ± 0.1	3.0 ± 0.4 **	3.7 ± 0.6 **	3.8 ± 1.1 **	7.1 ± 2.1 **	2.7 ± 0.8	2.8 ± 0.5 *	0.2 ± 0.1	<0.001
Monocytes	7.7 ± 2.8	7.3 ± 2.1	8.5 ± 1.3	10.8 ± 2.2	9.7 ± 2.9	9.0 ± 2.3	7.5 ± 1.6	7.4 ± 1.4	<0.001
Metamyelocytes	0	1.2 ± 0.3 **	3.7 ± 0.9 **	3.8 ± 1.0 **	6.2 ± 1.4 **	6.3 ± 1.5 **	7.5 ± 1.9 **	7.3 ± 2.0 **	<0.001
Band neutrophils	7.7 ± 3.0	12.8 ± 3.4 **	14.9 ± 2.5 **	20.3 ± 3.4 **	12.4 ± 3.0	4.5 ± 1.1 *	5.6 ± 1.2	5.6 ± 1.8	<0.001
Segmented neutrophils	23.1 ± 5.5	19.5 ± 4.0 *	5.3 ± 0.9 *	8.9 ± 1.5 *	3.5 ± 0.9 *	3.6 ± 0.9 *	3.7 ± 0.8 *	1.9 ± 0.3 *	<0.001
Eosinophils	4.9 ± 1.1	4.9 ± 1.3	1.6 ± 0.2 *	3.8 ± 0.8	1.8 ± 0.5 *	3.6 ± 0.7	2.8 ± 0.6	1.9 ± 0.2 *	<0.001
Basophils	0.2 ± 0.01	0.3 ± 0.02	0.4 ± 0.02	0.1 ± 0.02	0.3 ± 0.1	0.5 ± 0.1	0.4 ± 0.1	0.4 ± 0.1	<0.001
Plasmocytes	0	0	1.6 ± 0.3 **	3.2 ± 0.5 **	2.7 ± 0.7 **	3.6 ± 0.9 **	4.7 ± 0.9 **	4.6 ± 1.1 **	<0.001
Dead cells	0	0.6 ± 0.1	5.3 ± 1.0 **	8.9 ± 1.6 **	16.8 ± 3.4 **	16.2 ± 4.0 **	16.8 ± 3.8 **	24.8 ± 4.8 **	<0.001

Control represents the mean of a given cell type for each time point of infection. * Significant decrease compared with control (*p* < 0.05–*p* < 0.001). ** Significant increase compared with control (*p* < 0.05–*p* < 0.001).

**Table 3 life-12-01255-t003:** Cellular factors involved in ASFV entry.

Cellular Factors	Inhibitors	Reference
Na^+^/H^+^ channels	EIPA	[70]
Actin	Cytochalasin D and B	[71]
	Latrunculin A	
Myosin II	Blebbistatin	[71]
EGFR	324,674	[71]
PI3K	LY294002Worthmanin	[66]
Rac1	NSC23766, Rac1-N17	[72]
Pak1	IPA-3, Pak1-AID	[70]
Tyrosin kinases	Genistein	[71]
Dynamin-2	Dynasore	[66]
Clathrin	Clorpromazine	[71]
Microtubules	Nocodazol	[71]
Vacuolar acidification	Cloroquine, NH_4_Cl,	[66]
	Bafilomycin A	
Cholesterol	MβCD	[73]
Rab-7	Rab-7-T22N	[74]
	Rab-7 siRNA	
CD163	CD163siRNA	[70]
CD45	CD45siRNA	[73]
CD203a	CD203asiRNA	[74]
CD163	CD163siRNA	[70]

**Table 4 life-12-01255-t004:** ASFV genes and their functions.

Gene Name	Functional Group	Reference
A104R	Genome Organization	[102]
A151R	Structural/viral morphology	[103]
A179L	Immune evasion	[104]
A238L	Immune evasion	[105]
A489R	MGF 505	[106]
A505R	MGF 505	[107]
A506R	MGF 505	[107]
A528R	MGF 505	[106]
A542R	MGF 505	[108]
B119L	Other/enzyme	[109]
B125R	Transcription/RNA modification	[110]
B263R	Transcription/RNA modification	[111]
B318L	Other/enzyme	[112]
B438L	Structural/viral morphology	[42]
B602L	Structural/viral morphology	[113]
B646L	Structural/viral morphology	[114]
B962L	NA metabolism/DNA replication/repair	[115]
C257L	TR/PSP	[116]
C962R	NA metabolism/DNA replication/repair	[117]
CP204L	Other/enzyme	[118]
CP2475L	Structural/viral morphology	[45]
CP530R	Structural/viral morphology	[119]
D1133L	Transcription/RNA modification	[120]
D117L	Structural/viral morphology	[121]
D250R	Transcription/RNA modification	[122]
D339L	Transcription/RNA modification	[120]
E165R	NA metabolism/DNA replication/repair	[123]
E183L	Structural/viral morphology	[124]
E184L	TR/PSP	[125]
E199L	Structural/viral morphology	[126]
EP153R	Immune evasion	[127]
EP402R	Immune evasion	[128]
I196L	TR/PSP	[129]
K78R	Structural/viral morphology	[130]
L57L	Uncharacterized	[42]
M448R	Other/enzyme	[131]
O61R	Structural/viral morphology	[116]
QP383R	Other/enzyme	[132]
A240L	Thymidylate kinase	[133]
K196R	Thymidine kinase	[134]
F334L	Ribonucleotide reductase (small subunit)	[44]
F778R	Ribonucleotide reductase (large subunit)	[44]
G1211R	DNA polymerase family B	[44]
P1192R	DNA topoisomerase type II	[135]
E301R	Proliferating cell nuclear antigen (PCNA)-like	[44]
O174L	DNA polymerase X-like	[136]
NP419L	DNA ligase	[137]
E296R	AP endonuclease class II	[138]
EP1242L	RNA polymerase subunit 2	[139]
C147L	RNA polymerase subunit 6	[108]
NP1450L	RNA polymerase subunit 1	[139]
H359L	RNA polymerase subunit 3	[44]
D205R	RNA polymerase subunit 5	[44]
CP80R	RNA polymerase subunit 10	[44]
C315R	TFIIB-like	[140]
A859L	Helicase superfamily II	[141]
F1055L	Helicase superfamily II	[44]
D1133L	Helicase superfamily II	[120]
Q706L	Helicase superfamily II	[142]
QP509L	Helicase superfamily II	[143]
I243L	Transcription factor SII	[144]
NP868R	Guanylyl transferase	[145]
C475L	PolyA polymerase large subunit	[44]
EP424R	FTS J-like methyl transferase domain	[44]
EP364R	ERCC4 nuclease domain	[146]
D345L	Lambda-like exonuclease	[44]
B385R	VV A2L-like transcription factor	[44]
G1340L	VV A8L-like transcription factor	[147]
B175L	VV VLTF2-like late transcription factor, FCS-like finger	[44]
C962R	DNA primase	[117]
R298L	Serine protein kinase	[148]
I215L	Ubiquitin conjugating enzyme	[149]
D250R	Nudix hydrolase	[122]
A224L	IAP apoptosis inhibitor	[90]
DP71L	Similar to HSV ICP34.5 neurovirulence factor	[150]
KP177R	P22	[151]
A137R	P11.5	[152]
A78R	P10	[44]
B646L	P72 major capsid protein; involved in virus entry	[153]
B438L	P49; required for formation of vertices in icosahedral capsid	[154]
B602L	Chaperone; involved in folding of capsid; not incorporated into virions	[155]
B119L	ERV 1-like; involved in redox metabolism	[156]
S273R	SUMO-1-like protease; involved in polyprotein cleavage	[157]
CP2475L	pp220 polyprotein precursor of p150, p37, p14, and p34; required for packaging of nucleoprotein core	[45]
H108R	J5R; transmembrane domain	[158]
E120R	P14.5; DNA-binding; required for movement of virions to plasma membrane	[159]
E248R	E248R (k2R); possible component of redox pathway required disulphide bond formation	[160]
MGF 110-4L (XP124L)	XP124L; multigene family 110 member	[44]

**Table 5 life-12-01255-t005:** Host immune responses known to be regulated by African swine fever virus (ASFV).

Immune Response	Viral Genes	Immune Elements and Mechanisms	Impact on Virulence	Reference
Type I interferon response	A276R	Dampening type I IFN response by regulating IRF3	ND	[163]
Inflammatory response	A528R	Promoting the expression of ULK1 to degrade STING	Attenuated	[163]
	MGF360-12L	Interacting with nuclear transport proteins importin α (KPNA2, KPNA3, and KPNA4) to disrupt NF-κB nuclear translocation	ND	[171]
	I329L	Inhibiting the crucial adaptor protein TRIF	ND	[172]
	DP96R	Degradation of TBK1	Attenuated	[164]
	L83L	No reduction in virulence	No reduction in virulence	[167]
	A238L	Inhibiting the activation of the NF-κB pathway	No reduction in virulence	[161]
Apoptosis	226L, A151R, NP419L, QP383R	ND	ND	[137]
	A179L	Bind to pro-apoptotic proteins (Bid, Bim, Bak, and Bax) to inhibit apoptosis	ND	[104]
	A224L	Activating NF-κB pathway to promote anti-apoptotic genes expression, e.g., IAP and Bcl-2 family proteins	No reduction in virulence	[173]
	EP153R	Inhibiting the expression of caspase-3	ND	[174]
	E183L	Interacting with DLC8 to activate caspase-3 and caspase-9	ND	[175]
	DP71L	Recruit host phosphatase 1 (PP1) and remove the phosphorylation of eIF-2α to restore cellular protein synthesis to block CHOP activation suppressing apoptosis	No reduction in virulence	[176]
	A238L	Inhibiting CaN to decrease apoptosis; NF-kappaB antiapoptosis: induction of TRAF1 and TRAF2 and c-IAP1 and c-IAP2 to suppress caspase-8 activation	No reduction in virulence	[177]

**Table 6 life-12-01255-t006:** Resistance of ASFV to physical and chemical action [60].

Action	Resistance
Temperature	Highly resistant to low temperatures. Heat-inactivated by 56 °C/70 min; 60 °C/20 min.
pH	Inactivated by pH < 3.9 or >11.5 in serum-free medium. Serum increases the resistance of the virus, e.g., at pH 13.4 resistance lasts up to 21 h without serum, and 7 days with serum.
Chemicals/disinfectants	Susceptible to ether and chloroform. Inactivated by 8/1000 sodium hydroxide (30 min), hypochlorites as 2.3% chlorine (3 min), 3/1000 formalin (30 min), 3% orthophenylphenol (30 min) and iodine compounds.
Survival	Remains viable for long periods in blood, feces, and tissues, especially infected uncooked or undercooked pork products. Can multiply in vectors (*Ornithodoros* sp.).

**Table 7 life-12-01255-t007:** The mechanisms of action of disinfectants.

Aldehydes	Mutually Bind to Proteins, Inhibit Transport Mechanisms
Halogens (hypochlorite, iodophors, ClO_2_)	Penetrates the membrane and oxidizes proteins, interrupts the cell’s oxidative phosphorylation
Peroxides	Penetrates the membrane and oxidizes lipids, proteins, and DNA
Phenolics	Poisons the protoplasm and damages the cellular membrane
Quaternary ammonium compounds (QUATs)	Damages the cellular membrane and disrupts the membrane, cytoplasmic potential, and pH gradient

**Table 12 life-12-01255-t012:** Serogroup classification of African swine fever virus (ASFV) strains, isolates, and variants [278].

Serogroup	ASFV Reference Strain	ASFV Strains and Isolates	
		Highly/Moderately Virulent	Low Virulent Attenuated/Avirulent
1	Lisbon-57	Lisbon-57 (L-57), Kimakia, Katanga-78, Katanga-105, Katanga-115, Madeira-65, Diamant	LS, L-50, LF-97, Kimakia-155, Diamant-160, Katanga-139 (Kc-139), Katanga-149 (Kc-149), Katanga-160 (Kc-160), LK-111, Katanga-350
2	Congo-49	Congo-49 (K-49), Yamba-74, Le Bry-73, Sylva	NVL-1, Mfuati-79, Ndjassi-77, KK-202, KK-262/C
3	Mozambique-78	Mozambique-78 (M-78), MK-101	MK-200, MK-210
4	France-32	France-32 (F-32), Cuba-71, Brazil-80, Cuba-80, Malta-78, Sao-Tome, and Principe-79 (STP-1), DNOPA-Luanda, Odessa-77	FK-32/135
5	TSP-80	TSP-80	TSP-80/300
6	TS-7	TS-7	TS-7/150, TS-7/230
7	Uganda	Uganda	UK-50, UK-80
8	Rhodesia, Stavropol 01/08	Rhodesia, Stavropol 01/08	St-CV1/20
9	Davis	Davis	None

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
