# Peer review of "African Swine Fever Virus: A Review"

_life, 2022, doi:10.3390/life12081255_

Round 1
Reviewer 1 Report
This review describes the current state of art of different aspects of ASF and ASFV. It includes all relevant points and data. There are already several published reviews with similar information. Therefore, in order to publish a novel approach about this topic, I suggest to make this article easier to read by implementing English correction and a simpler approach in table form. Sections 2 to 7 should sum up the information in tables gathering all relevant information and literature sources.
Some minor points need also revision:
Introduction
L35: “despite the launch of emergency measures by the Chinese government” please revise that sentence, verb is missing
L36-38: “Domestic pigs infected with ASFV may present a variety of clinical symptoms and differences among virus strains and host susceptibilities can lead to different clinical responses” revise English writing and describe a bit more accurately what the differences are.
L38-39: add a reference for this statement.
L40-42: ASFV is not as easy to transmit as stated here. Oral infection not always works and the best infection route is through ASFV-contaminated blood. I suggest authors to develop further this statement and add the relevant literature, as well as relevant studies where oral-infection did not work as planned.
L42-44: add literature about the economical impact of ASF.
L42-46: in Vietnam a novel vaccine has been licensed, add the relevant information.
2. African Swine Fever
L48: rephrase this statement. It implies that ASFV could cause more than one disease
L49-52: rephrase this statement. What do you mean with the mean route of transmission?
L52-53: Replication in ticks has only been shown for a few species and viral strains, I consider to expand this part and add the relevant literature.
L58: what form?
L54-75: replace this paragraph by a relevant table with the percentage of animals and/or related strains found in the literature that could course “peracute, acute, subacute, or chronic”. Related symptoms, age of animals and literature should be added in this table.
L68: “Pregnant sows may miscarry” add reference
L75: add the genotype
L83-87: add relevant literature
3. ASFV Structure
L108: could you develop this statement and highlight the possible explanation for the bigger viral particle found in these Chinese farms?
L109-111: please, look for synonyms of the word “contains”, it has been too often used in these sentences.
In this section, relevant information about the variations found in Poland in the Polymerase X are missing. I suggest authors to expand this part and discuss the variants that might have appeared in Poland and Germany as a result of the variability of this enzyme.
L122: expand this emersion mechanisms and state briefly what’s known about it
4. Virus replication
L182: What diversity of pathways? Please, rephrase this statement
For this part of the review, I would also suggest to make a table summing up all known receptors, their possible role in virus replication and the relevant literature where it can be found.
Would it be possible to find a figure that describes the virus replication? It would be more helpful to describe this part of the paper referring to a figure that could show the different steps. If this was possible, I would suggest to replace figure 2.
5. Virulence Gene
Correct it for “Genes”
L253: ASFV causes
For this part of the review, I would also suggest to make a table summing up all known genes, their possible role in virulence and the relevant literature where it can be found. Including the isolates and genotypes known for this virulence.
6. Immunoscape
For this part of the review, I would also suggest to make a table summing up all known host immune mechanisms, their possible role in virulence and the relevant literature where it can be found. It would be also desirable to briefly discuss all in the context of vaccine development and understanding correlates of protection.
7. Virus Inactivation
Please add more detailed information about ASFV inactivation with non-ionic chemical agents.
8. Virus Typing
Needs English proof.
9. Disease Dissemination and Control
Would it be possible to add a link reference to wahis or similar where the reader will find all the updated cases?
L586-587: add the updated information about the licensed vaccine.
L585-608: before discussing the improvements that should be made in order to control ASF, a small introducing paragraph should be added where the current state of the diagnostics and control by the government is. Then, I suggest to split the text from the discussion stated here by the authors.
There is a general lack of references in this part of the review.
Needs English proof.
10. Vaccination
Please, update accordingly.
Reviewer 2 Report
Article entitled “African Swine Fever Virus: A mini Review” has some scientific value. Reviewed article requires some corrections and additions.
1. The authors do not mention the mechanisms of pathology at the organismal level.
2. Statement that at autopsy the most characteristic lesion of acute ASF is hemorrhagic splenomegaly incorrect (there are several another symptoms such as hemorrhages, affected lymph nodes, ets.).
3. Statement that “subacute ASF shows similar symptoms to the clinical symptoms observed in acute ASF, although these are generally less pronounced” also need a serious corrections.
4. Changes in the content of leukocytes in the peripheral blood are described very briefly and do not reflect the overall picture of the pathology (significant additions are needed in this section, in particular in the pathology caused by the genotype II virus).
5. Statement that “while the red blood count remains almost unchanged” belong only to the some genotypes. These data should be supplemented with new data.
6. Changes in the neutrophil population, both quantitative and qualitative, are described only as neutrophilia, which is not enough. The role of diapedesis should be included.
7. Data on the role of the Hemadsorption (HAD) assay in ASFV diagnosis should be added.
Minor remarks:
8. What means “strongly virulent America /07 ASFV” at line 342? May be Armenia?
9. Usually Vero cell written with capital letter (line 181).
10. Lines 468-469. First sentence “Adding serum to a serum-free medium can increase the resistance of the virus” Second sentence “For example, at pH 13.4, resistance can last 21 h without serum and 7 days with serum”. Something wrong
11. Lines 493-494 “ASFV can be inactivated in serum-free media at pH < 3.9 or pH > 13.4”. Looking like incomplete sentence.
Round 2
Reviewer 1 Report
Introduction
L44: hyperacute
L 58- 61: This sentence needs English proofreading
L 110: “Clinical signs and lesions are induced…”
7. Virus Inactivation
Please add the following reference https://www.mdpi.com/2076-0817/11/7/750
Author Response
Dear reviewers,
Thank you for your valuable suggestions. We have made modifications according to your requirements. The detail of the main corrections in the paper are as followings:
L44: "peracute (or yperacute)" was corrected as "hyperacute".
L58-61: The sentence has been revised.
L110: "This form is" was modified to "Clinical Signs and Lesions are".
The following reference was added at L663.
Thank you once again for your attention to our paper. We look forward to your reply.
Best Regards.
Yours sincerely,
Wenxian Chen
Reviewer 2 Report
The authors have qualitatively changed the work and it generally corresponds to the journal.
There are only a few minor remarks
There is no ArmeniaAmerica /07 strain but just Armenia07
The data of pathoanatomical studies, if possible, should be presented in the article in a larger illustration
Author Response
Dear reviewer,
Thank you for an affirmation of our work. We have made modifications according to your requirements.
Thank you once again for your attention to our paper. We look forward to your reply.
Best Regards.
Yours sincerely,
Wenxian Chen